# High spatiotemporal resolution traffic CO₂ emission maps derived from Floating Car Data (FCD) for 20 European cities (2023)

Qinren Shi[1], Philippe Ciais[1], Nicolas Megel[2], Xavier Bonnemaizon[1], Rohith Teja Mittakola[1], Richard Engelen[3], Chuanlong Zhou[1]

[1]Le Laboratoire des Sciences du Climat et de l'Environnement, Saint-Aubin, 91190, France
[2]NEXQT SAS, Paris, France
[3]ECMWF, Robert-Schuman-Platz 3, 53175 Bonn, Germany

*Correspondence to*:  Qinren Shi(qinrenshi2025@gmail.com)  & Chuanlong Zhou(chuanlong.zhou@lsce.ipsl.fr)

**Abstract.** On-road transportation is a major contributor to $CO_2$ emissions in cities, and high-resolution $CO_2$ traffic emission maps are essential for analyzing emission patterns and characteristics. In this study, we developed new hourly on-road $CO_2$ emission maps with a $100 \times 100$ m resolution for 20 major cities in France, Germany, and the Netherlands in 2023.  We used commercial Floating Car Data (FCD) based on anonymized GPS signals periodically reported by individual vehicles, providing hourly information on mean speed and the number of GPS sample counts per street. Machine learning models were developed to fill FCD data gaps and convert sample counts into actual traffic volumes, and the COPERT model was used to estimate speed- and vehicle-type-dependent emission factors. These models were calibrated using independent traffic observations available for Paris and Berlin, and subsequently applied to the remaining 18 cities in an extrapolated manner due to data availability constraints. Hourly emissions, initially estimated at the street level, were aggregated to $100 \times 100$ m grid cells. Annual on-road $CO_2$ emissions across the 20 European cities in 2023 ranged from 0.4 to 7.9 Mt $CO_2$, with emissions strongly correlated with urban area ($R^2 = 0.98$) and, to a lesser extent, population size ($R^2 = 0.74$). Spatially, emissions are either highly concentrated along major highways in cities such as Paris and Amsterdam or more evenly distributed in cities such as Berlin and Bordeaux, highlighting the need for context-specific mitigation strategies. Temporally, this study shows the $CO_2$ emission fluctuations due to holiday periods, weekly activity cycles, and distinct usage profiles of different vehicle types. Due to the low latency of FCD, this approach could support near-real-time traffic emission mapping in the future. Our approach enhances the spatial and temporal characterization of $CO_2$ emissions in on-road transportation compared to the conventional method used in gridded inventories, indicating the potential of FCD data for near-real-time urban emission monitoring and timely policy-making. The datasets generated by this study are available on Zenodo https://doi.org/10.5281/zenodo.16600210(Shi et al., 2025).

## 1 Introduction

The road transport sector is one of the largest sources of greenhouse gas (GHG) emissions in the European Union and the only major economic sector where carbon dioxide ($CO_2$) emissions have risen since 1990, primarily due to the widespread use of fossil fuel-powered passenger cars and freight vehicles. In 2023, it accounts for approximately 26.0% of total EU GHG emissions (EEA, 2024a). In response to the dual challenge of reducing emissions and developing cleaner mobility infrastructures, the European Strategy for Low-Emission Mobility outlines three elements: (1) Increasing the efficiency of the transport system, including the optimization of logistics and intelligent transport systems; (2) Accelerating the deployment of low-emission alternative energy sources, such as biofuels, renewable electricity, and hydrogen; and (3) Speeding up the transition to zero-emission vehicles, through regulatory incentives, infrastructure investment, and innovation support (European Commission, 2016). This transition is not only critical for achieving the EU's climate neutrality goal, which involves reducing net $CO_2$ emissions to zero by 2050 (EEA, 2024b), but also for improving air quality, reducing energy dependence on fossil fuel imports, and enhancing the competitiveness of European industry.

Emission reduction targets in the transportation sector are being translated into concrete actions at the city level. For instance, the transportation sector is responsible for approximately 20% of Paris' local greenhouse gas emissions (Albarus et al., 2025), and Paris plans to reduce its direct emissions by 50% by 2030 and 100% by 2050, compared to 2004. Paris has set itself the target of phasing out diesel-powered mobility by 2024 and petrol-powered mobility by 2030, aligning with the EU-wide ban on the sale of internal combustion engine vehicles by 2035. Amsterdam aims to achieve zero-emission transport by 2030, phasing out all fossil-fuel vehicles within city limits (Amsterdam, 2024). The city is rapidly expanding its electric vehicle infrastructure, as all newly registered vehicles are required to have zero-emission engines in 2025 (CINEA, 2025). Similarly, to achieve climate neutrality in 2050, Berlin will require a long-term reduction in $CO_2$ emissions in the transport sector to around 1.17 million tonnes of $CO_2$ per year, a reduction of around 77 % compared with 1990 emissions (diBEK, 2025).

High-resolution emission maps are crucial for monitoring emission changes and providing insights into the effectiveness of traffic mitigation policies in cities. For example, a high-resolution (1 km²) $CO_2$ emissions inventory for U.S. road transportation named DARTE enables detailed analysis at the city scale between 1980 to 2012 (Gately et al., 2015), revealing that urban areas drive most of the emission growth and that traditional population-based downscaling methods substantially misrepresent city-level spatial patterns. Over the past decade, several efforts have been made to improve either the temporal or the spatial resolution of traffic emission inventories, primarily by incorporating real-world traffic data generated from sensors or GPS signals. From a temporal resolution perspective, annual aggregated statistics make it impossible to capture short-term variations due to weather, policy changes, or special events. Therefore, daily or hourly data were increasingly applied to improve the accuracy. For example, TomTom collects all the travel times and compares them with the lowest travel times to calculate congestion indexes based on FCD (index, 2024). Tomtom congestion indexes were used by Carbon Monitor Cities

(Huo et al., 2022) to estimate daily $CO_2$ emissions for 1500 cities. CAMS-TEMPO is a dataset of European emission temporal profiles that provides gridded monthly, daily, weekly, and hourly weight factors for atmospheric chemistry modelling, and the European part used hourly traffic data collected from over 20 European cities via open-data portals or personal communications (Guevara et al., 2021). One-month GPS-based datasets covering 52,834 conventional fuel vehicles registered in the province of Modena and 40,459 vehicles registered in the province of Firenze were used to generate high-resolution emission maps (De Gennaro et al., 2016). A near-real-time on-road traffic emission product on 2860 km of the main roads in Bangkok was automatically generated by retrieving the traffic data from the Google Maps API service and the Python code every 15 min (Naiudomthum et al., 2022). In recent years, machine learning-based bottom-up approaches have supported the development of high-resolution emission maps. For instance, an hourly street-level emission map of Chengdu was developed using data from 1,454 camera-based sensors and 34 highway monitoring sites, employing land-use random forest models (Wen et al., 2022). Similarly, a platform tracking hourly $CO_2$ emissions at a 30×30 m resolution was designed for Berlin based on local traffic data, using machine learning methods (Anjos and Meier, 2025).

Despite recent advancements, most city-level emission datasets still suffer from limitations in either temporal or spatial resolution, with few achieving both simultaneously. CAMS-TEMPO (Guevara et al., 2021) and Carbon Monitor (Huo et al., 2022) lack road-specific information and provide only outputs at 0.1° resolution and the city level, respectively. The hourly street-level emission datasets for Chengdu (Wen et al., 2022) and Bangkok (Naiudomthum et al., 2022) only cover one to two months. The Berlin platform offers high spatial and temporal resolution from 2015 to 2022, but may miss data from smaller roads, as counting stations are usually located on major roads.

As part of the Copernicus Atmosphere Monitoring Service (CAMS), this study estimates for the first time hourly street-level on-road transportation $CO_2$ emissions, aggregated into 100 m resolution hourly maps for 20 European cities in 2023. Hourly GPS-based data, reporting traffic counts and speeds of individual vehicles across different road classes, were upscaled using machine learning to reconstruct complete traffic volumes and speeds across the road networks. Then, $CO_2$ emissions were estimated using the COPERT model, and emission maps were developed. This approach enhances the spatial and temporal characterization of $CO_2$ emissions in on-road transportation compared to the downscaling method used in other inventories, indicating the potential of GPS-based data for supporting future efforts in emission monitoring and developing emission reduction policies.

**2 Data and Method**

**2.1 Overview of the Methodology**

Figure 1 describes the workflow of this study. The GPS-based high-resolution 'Floating Car Data' (FCD) on individual vehicle flow (GPS vehicles counts per street each hour) and speed covering every street was obtained from a data aggregation provider

that collects GPS position data from cars (passenger cars) and trucks (light commercial vehicles and heavy duty trucks), providing road-specific information on hourly average speed and sample counts (i.e., the number of cars recorded in each street for each hour). Those GPS data are linked with precise cities' road network datasets, providing detailed information on road length, road functional class, and truck access authorization. All data is anonymized by the data provider to prevent compromising any individual or organizational data privacy issues. After raw data processing and cleaning, a machine learning model was used to fill in missing values in FCD, as well as to transform FCD sample counts limited to vehicles equipped with GPS into traffic volumes for all vehicles. Then, the COPERT model (Ntziachristos et al., 2009), the EU standard vehicle emissions calculator, was applied for estimating specific $CO_2$ emission factors based on individual vehicle hourly average speed and type. Combined with the road lengths obtained from geographical databases and with fleet structures, we finally estimate street-level road-specific emissions using the following equation:

$$Emis_{t,v,r} = N_{t,r} \times Structure_v \times Length_r \times EF_{v,s} \tag{1}$$

Where $Emis_{t,v,r}$ represents $CO_2$ emission at the hour $t$, for the vehicle type $v$, on road $r$. $N_{t,r}$ represents the total traffic volume at hour t, on road r (counts/hour). $Structure_v$ represents the proportion of vehicle type $v$ in the vehicle fleet (%). $Length_r$ represents the road length (km) of the road r, and $EF_{v,s}$ (g $CO_2$/km) represents the $CO_2$ emission factors for the vehicle type $v$, at the hourly average speed $s$ (km/h).

Our FCD source covers France, Germany, and the Netherlands. Therefore, the 20 most populous cities within these three countries were selected to develop high-resolution emission maps. Table 1 shows the basic information (population, area, street length, street density) of the 20 cities in 2023. Note that here Paris is the administrative city jurisdiction (Ville de Paris) covering the central 20 arrondissements, so its area is much smaller than Berlin, which is both a city and a federal state.

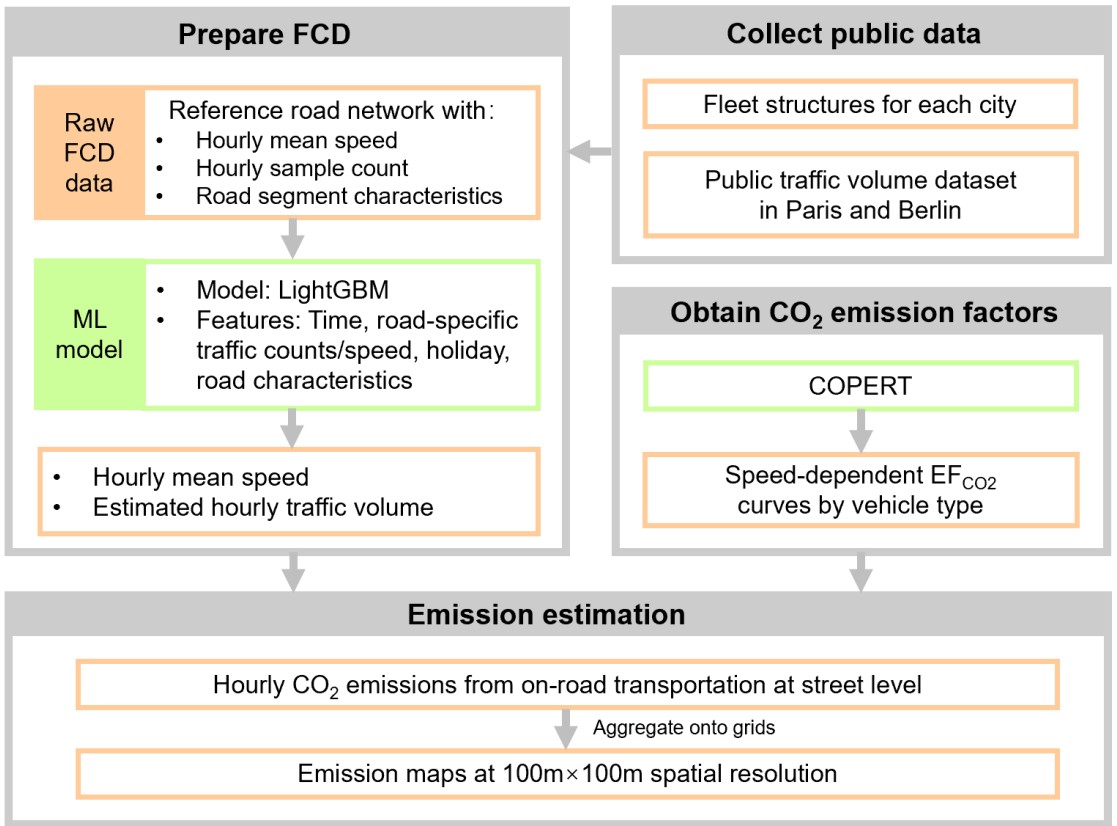


**Figure 1: Workflow of this study**


**Table 1: Information of 20 selected cities in 2023.**

| Country | City | Population (Thousand) | Area (km²) | Street length (km) | Street density (km/km²) |
|---|---|---|---|---|---|
| France | Paris | 2,103 | 105.4 | 2412.9 | 22.9 |
| | Marseille | 862 | 240.6 | 3301.7 | 13.7 |
| | Lyon | 513 | 47.9 | 985.3 | 20.6 |
| | Lille | 233 | 39.5 | 679.8 | 17.2 |
| | Toulouse | 472 | 118.3 | 2311.2 | 19.5 |
| | Nice | 343 | 71.9 | 1228.0 | 17.1 |
| | Nantes | 303 | 65.2 | 1249.4 | 19.2 |
| | Strasbourg | 277 | 78.3 | 1252.4 | 16.0 |
| | Montpellier | 278 | 56.9 | 1260.1 | 22.1 |
| | Bordeaux | 250 | 49.4 | 967.9 | 19.6 |
| Germany | Berlin | 3,782 | 891.3 | 12073.4 | 13.5 |
| | Hamburg | 1,910 | 755.2 | 8725.2 | 11.6 |
| | Munich | 1,510 | 310.7 | 5220.0 | 16.8 |
| | Cologne | 1,087 | 405.2 | 5508.8 | 13.6 |
| | Frankfurt | 776 | 248.3 | 3648.5 | 14.7 |
| | Stuttgart | 633 | 207.3 | 3660.8 | 17.7 |
| | Dusseldorf | 631 | 217.4 | 2741.5 | 12.6 |
| Netherland | Amsterdam | 883 | 219.4 | 3203.8 | 14.6 |
| | Rotterdam | 656 | 324.1 | 3555.7 | 11.0 |
| | The Hague | 553 | 98.1 | 1796.8 | 18.3 |


**2.2 Description and preparation of FCD**
FCD provides hourly average speed and sample counts for each street, with separate data for cars and trucks reporting GPS
data. The FCD is linked with high-resolution road network datasets that feature information such as road length, speed category,
road functional class, lane category, on more detailed and complete road networks than public traffic datasets based on sensors.
As shown in Figure 2, public datasets used by previous studies are only available for a few cities and provide hourly traffic
data for 3,739 road segments in Paris (Xavier Bonnemaizon 2024) and 19,808 segments in Berlin (Anjos and Meier, 2025),
respectively. In contrast, FCD gives vehicle count samples and speed information for 36,716 roads in Paris and 122,759 roads
in Berlin, dividing long roads into more segments and encompassing a much greater number of small roads than the city-level
public datasets. All road segments were categorized into major, middle, and small according to the functional class defined by
the FCD. Major roads represent roads connecting major metropolitan areas, middle roads represent roads connecting
neighbourhoods, and small roads represent low-volume roads.

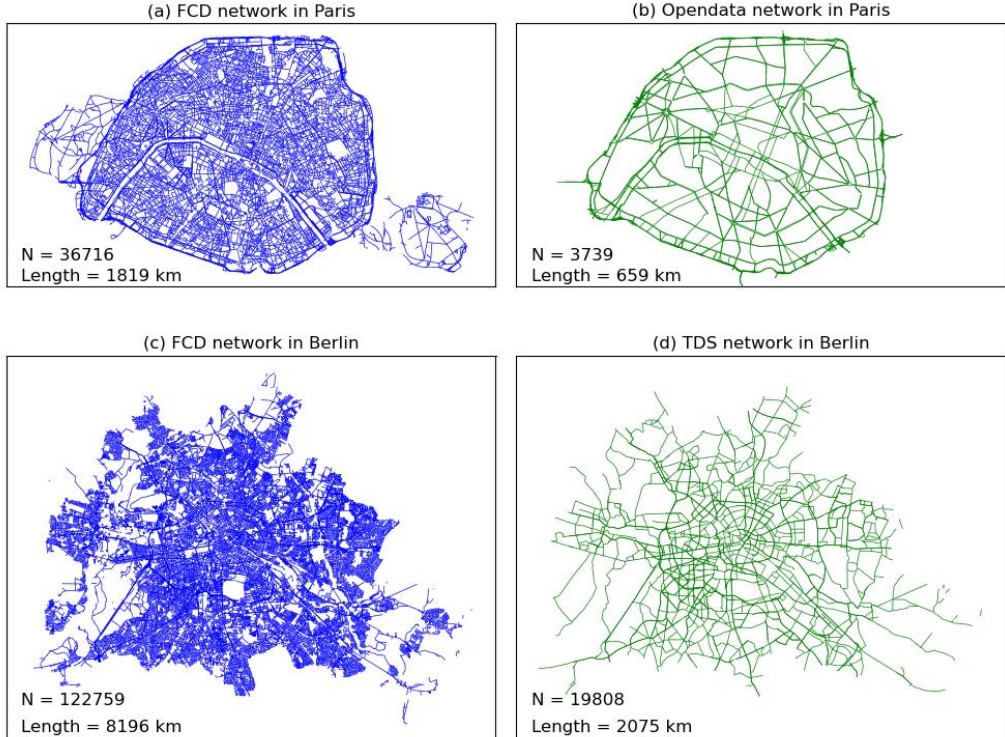


**Figure 2: Monitored road networks in this study and other public datasets in Paris and Berlin.** N represents the number of road
segments. (a) and (c) represent road networks from FCD for Paris and Berlin, respectively; (b) and (d) represent networks from Open Data
in Paris and Traffic Detection Systems in Berlin.

Missing values exist in the FCD due to unstable GPS signals, especially for small roads. The average data coverage of GPS
cars on major, middle, and small roads ranges from 67.0% - 97.7%, 40.4% - 93.8%, and 6.1% - 37.7%, respectively (Figure
S1a). The average data coverage of trucks is lower, ranging from 32.2% - 75.8%, 32.1% - 85.3%, and 1.8% - 32.2%,
respectively (Figure S1b). Machine learning was used here to fill data gaps, as the use of machine learning techniques has
shown great potential for both temporal and spatial imputation of missing data to reconstruct the full volume of traffic(Wen et

al., 2022). Eight features were chosen as predictors (Table 2) to train models. Temporal features (hour, day of the week, and month) were used to capture diurnal and seasonal patterns in traffic behaviour. Observed road-specific daily mean traffic counts and speeds derived from hourly averages were also used as indicators of baseline traffic intensity. Holiday indicators, including school and public holidays, were included to account for potential shifts in travel demand. Finally, road characteristics including speed category, functional class, and lane category were used to describe the physical and functional attributes of each road segment.

**Table 2: Spatial-temporal features used as predictors of traffic variables**

| Category | Features | Usage |
|---|---|---|
| Time | Hour, Day of week, Month | Diel and seasonal pattern |
| Road-specific traffic counts/speed | Daily mean derived from hourly averages | Baseline traffic intensity |
| Holiday | School holiday, Public holiday | Potential shifts in travel demand |
| Road characteristics | Speed category, Functional class, Lane category | Road capacity and flow characteristics |

The full-year dataset was partitioned into two temporally isolated subsets: January-June (H1) and July-December (H2) due to the large-scale dataset. Separate machine learning models were developed for each six-month interval, both incorporating consistent feature engineering protocols for vehicle type differentiation (Cars and Trucks) and road classification. Model training was conducted on 80% of the available data, with the remaining 20% held out as an independent test set to evaluate generalization performance. Random forest (RF) and lightGBM models were tested for Paris to compare their performances. As shown in Table S1, Random Forest (RF) and LightGBM exhibited comparable predictive performance across different vehicle types, road types, and target variables (i.e., vehicle count and speed) but LightGBM required significantly less

computational time. In some cases, the efficiency gain is more than 10-fold e.g., to fill gaps of car count on major roads takes 6.25 s for LightGBM vs. 122.53 s for RF. This efficiency gain stems from LightGBM's histogram-based decision tree learning and its leaf-wise tree growth strategy with depth constraints, which together enable faster training and better scalability, especially for large datasets with continuous features. Given its high accuracy and computational efficiency, LightGBM was chosen as the preferred model and trained individually for each of the 20 cities.

The LightGBM validation performance is summarized in Table 3 using mean $R^2$, RMSE, and MAE across cities and road classes, while the full city-level validation results are reported in Table S2. 5-fold cross-validation results which aimed at evaluating the robustness of the model are presented in Table S3. Overall, the model demonstrates strong predictive performance across different vehicle types and target variables. For car count, performance is consistently high on major roads, with $R^2$ values typically above 0.90 and reaching up to 0.97 (e.g., The Hague and Amsterdam). On middle and small roads, $R^2$ varies between 0.53 and 0.85, and lower values are often observed in cities with smaller datasets, such as Lyon and Nice, suggesting that data volume plays a critical role in model accuracy (Figure S2). For car speed, the model also performs well on major roads $R^2$ (0.85-0.95) but shows greater variability on smaller roads, where $R^2$ drops to as low as 0.39 in some cases (e.g., Paris or Lyon). The results of trucks are similar to those of cars, but with slightly lower overall performance. Shapley values, a concept from cooperative game theory, are widely used to explain feature importance in machine learning. This study used the Python package SHAP to estimate Shapley values applied to the model's conditional expectation function (SHAP, 2025), revealing that the daily mean count and hour of day are the most influential predictors, followed by day of week, road class, and month (Figure S3). High traffic volumes are associated with increased model output, while hourly effects vary by time of day. In contrast, features such as lane type and school holidays show limited influence.

**Table 3: Summary of LightGBM validation performance across cities and road classes.**

| Vehicle | Item | Road class | Mean R² | Mean RMSE | Mean MAE |
|---------|------|-----------|---------|-----------|----------|
| Car | COUNT | Major | 0.93 | 16.34 | 9.08 |
| Car | COUNT | Middle | 0.73 | 6.09 | 3.91 |
| Car | COUNT | Small | 0.60 | 3.66 | 2.15 |
| Truck | COUNT | Major | 0.78 | 3.31 | 2.00 |
| Truck | COUNT | Middle | 0.57 | 1.88 | 1.29 |
| Truck | COUNT | Small | 0.54 | 1.87 | 1.15 |
| Car | SPEED | Major | 0.89 | 6.72 | 4.64 |
| Car | SPEED | Middle | 0.67 | 6.71 | 4.87 |
| Car | SPEED | Small | 0.58 | 7.85 | 5.63 |
| Truck | SPEED | Major | 0.84 | 8.77 | 6.35 |
| Truck | SPEED | Middle | 0.55 | 7.81 | 5.85 |
| Truck | SPEED | Small | 0.56 | 7.70 | 5.65 |

## 2.3 Obtain $CO_2$ emission factors using COPERT

To calculate the speed-dependent emission factors $EF_{CO_2}$ defined by $CO_2$ emissions per km driven for each vehicle type, we applied the COPERT model, a widely used emissions calculator for vehicles in Europe (Ntziachristos et al., 2009). Monthly temperature and relative humidity data required as input for COPERT were obtained from ERA5 reanalysis (Hersbach, 2023) and interpolated to a 0.01° spatial resolution. City-level averages of maximum/minimum temperature and relative humidity were then calculated within administrative boundaries defined by Eurostat shapefiles to serve as inputs for COPERT. Considering the data scale and time cost, instead of running COPERT for each street segment each hour, this study developed fitting curves between speed and $EF_{CO_2}$ to obtain $EF_{CO_2}$. Except for L-Category vehicles running on diesel, where COPERT provides a fixed value, emission factors were simulated for various vehicle types at speeds of 20, 40, 60, 80, 100, 120, and 140 km/h. Then, for each city, cubic functions were fitted to COPERT simulations, as given by:

$$EF = a \times s^3 + b \times s^2 + c \times s + d \qquad (2)$$

Where s represents the average speed at hourly resolution, and $a$, $b$, $c$, and d are city-specific constants. Table S4 presents the parameters of the curve fitting results for all cities, showing a good fit quality with an R² value range from 0.882 to 0.998. In this way, the corresponding emission factor for any given speed can be determined. Note that we used $EF_{CO_2}$ of the EU6 standard, since $CO_2$ emission factors are only marginally influenced by emission standards, and this approach was also adopted by TomTom (Index, 2024).

**2.4 Estimate real traffic volume from sample count**

Road-specific hourly total traffic volume is the key parameter to estimate $CO_2$ emissions. Since not all vehicles transmit GPS signals and our dataset only captures a subset of the real GPS data for all vehicles, the actual traffic volume is significantly higher than the sample counts from the FCD. To solve this problem, we established a relationship between real traffic volume data and GPS sample count using machine learning. Due to the availability of traffic volume data, only the Opendata from Paris (Parisopendata, 2024) and Traffic detection Berlin (Berlinopendata, 2024) were used for modelling. Opendata from Paris provides hourly total vehicle flow from permanent sensors with electromagnetic loops on 2278 roads in 2023, but does not differentiate between vehicle types for the traffic volume. Therefore, the numbers of cars and trucks are estimated based on the proportion of sample counts from each type in our FCD. Traffic detection in Berlin provides hourly total vehicle volumes on 231 roads, and only the volumes of cars were used for modelling. As shown in Figure 2, monitored road networks of public datasets and FCD are different. The overlap rate and angle are used as criteria to link the two datasets' shapefiles (Figure S4). When the overlap rate > 0.7 and the angle <20°, a road is identified as being the same in Opendata and FCD. In this way, hourly open data from 2278 monitoring sites in Paris and 231 monitoring sites in Berlin were matched to the FCD, and we got the real volume and the number of FCD sample counts on the same road. A similar set of predictors as listed in Table 2, except for road-specific traffic counts and speeds, was used to build a LightGBM model to extrapolate FCD sample counts to total traffic volume. For cars in German cities, we used the LightGBM model trained on Berlin's data, while for all other cities, we used the LightGBM trained on Paris's data. The validation results (Table S5) show that the LightGBM model performs well on major roads in both Paris ($R^2$ = 0.91 for cars and 0.88 for trucks) and Berlin ($R^2$ =0.66 for cars). The accuracy decreases on middle and small roads in Paris ($R^2$ range from 0.22 to 0.38), while the performance in Berlin remains comparatively good ($R^2$ range from 0.86 to 0.88). 5-fold cross-validation results are presented in Table S6.

In addition to Paris and Berlin that are used for model training, observed traffic-count-based annual average daily traffic flow (AADT, in number of vehicles per day) or annual average weekday traffic (AAWT, equivalent to AADT excluding weekends) datasets are available for six additional cities reported in a recent study(Bonnemaizon et al., 2025): Montpellier and Hamburg (AADT), and Bordeaux, Lyon, Toulouse and Lille (AAWT). The comparison which serves as independent external validation to assess our traffic volume estimates is shown in Figure 3. Paris, the most important reference city for model development, shows strong agreement between estimated and public AADT values ($R^2$ = 0.92, n = 2696), with data points across all road classes closely aligned with the 1:1 line. Berlin exhibits noticeably larger dispersion, with a moderate $R^2$ (0.55) derived from a relatively small sample size (n = 197), which likely contributes to the lower correlation.

Lyon, Hamburg, Bordeaux and Montpellier all show moderate correlation (with $R^2$ around 0.6). However, while simulated and observed traffic volumes are generally well aligned for Bordeaux, public observations for Lyon, Hamburg and Montpellier tend to exceed the simulated values, especially for the major roads. Toulouse and Lille are characterized by low correlation

(R² around 0.3), exhibits the weakest consistency between estimated and public traffic volumes. Overall, the scatter plots reveal
pronounced city-to-city heterogeneity in traffic volume agreement, providing important context for subsequent uncertainty
propagation to city-scale emission estimates.

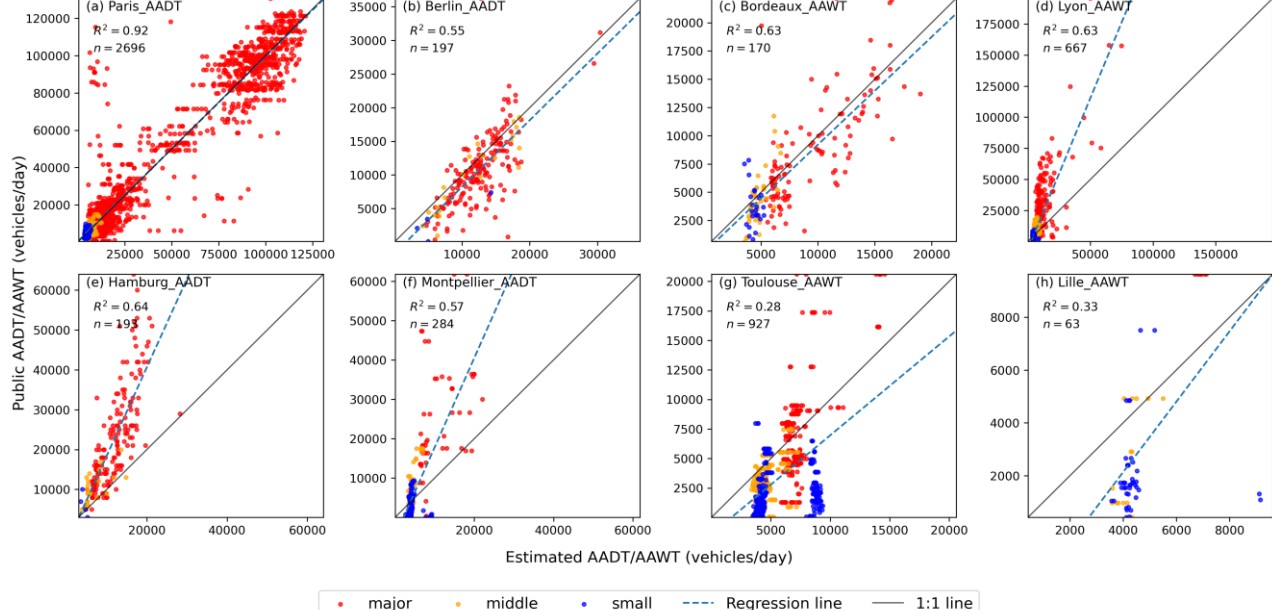


**Figure 3: Comparison of AADT/AAWT between this study and public datasets**

**2.5 Fleet structure**
This study collected fleet structures data in 2023 for the 20 cities to further map cars and trucks to 5 categories (passenger cars,
light commercial vehicles, buses, L-category and heavy-duty trucks), and 12 sub-categories, 10 fuels (petrol, diesel, CNG,
diesel hybrid, biodiesel, diesel PHEV, CNG biofuel, petrol hybrid, battery electric), as shown in Table 4. The data that is
reported annually was collected from the official statistical websites of France, Germany, and the Netherlands (Table S7).
Only direct emissions from fossil fuels are considered, so the emission factor of battery electric cars is set to 0.






**Table 4: Vehicle categories**

| Big Category | Category | Fuel |
|---|---|---|
| Car | L-Category | Petrol, Diesel |
| | Buses | Petrol, Diesel, CNG, Diesel Hybrid, Biodiesel, Battery electric, Diesel PHEV |
| | Passenger Cars | Petrol, Diesel, CNG, Petrol Hybrid, Petrol PHEV, Battery electric, Diesel PHEV |
| Truck | Heavy Duty Trucks | Petrol, Diesel, Diesel PHEV, Battery electric, CNG |
| | Light Commercial Vehicles | Diesel, Petrol, Diesel PHEV, Battery electric, CNG, Petrol Hybrid, Petrol PHEV |


## 2.6 Aggregation onto grids

Python was used to map street network emissions data onto a 100 × 100 m grid. Starting from a shapefile containing road
segments with associated emissions, a spatial join was performed using GeoPandas' sjoin function to identify which road
segments intersect each grid cell. Emissions were then allocated to the grid cells in a length-weighted manner, proportionally
distributing each road segment's emissions based on the length of its overlap with each cell. For the projections, cities in
France use EPSG:2154, while most German cities use EPSG:25832; Berlin uses EPSG:25833 due to its location. Dutch cities
are projected using EPSG:28992.

## 2.7 Uncertainty analysis

Monte Carlo method is widely used in emission studies to estimate uncertainties(Ramírez et al., 2008; Zhao et al., 2011; Super
et al., 2020). To quantify the uncertainty in estimated annual emissions arising from uncertainty in traffic volume estimates,
this study applied a Monte Carlo simulation framework that propagates the observed discrepancies between estimated traffic
volumes and public AADT/AAWT datasets (Figure 3) to the city-scale emission. Because emissions are linearly proportional
to traffic volume, uncertainty in traffic counts can be directly transferred to emission uncertainty. As standard parametric
assumptions (e.g., lognormality) did not adequately describe the tails of the discrepancy distributions, this study adopted a
fully empirical cumulative distribution function (ECDF) approach. Discrepancy ratios were grouped by functional road class
(major, middle, and small). For the six cities with observed AADT/AAWT data(Paris, Berlin, Bordeaux, Lyon, Hamburg,
Montpellier, Toulouse and Lille), discrepancy ratios were sampled directly from the city-specific ECDFs. For cities without
observations, we used country-level pools: ratios for French cities were sampled from the pool formed by the observed French

cities, ratios for German cities from the observed German cities, and ratios for Dutch cities from a combined pool of the observed French and German cities.

For each Monte Carlo iteration $j$, the set of ratio values corresponding to a given road class was selected. A random value $u \sim U(0,1)$ was drawn, and the corresponding correction factor was obtained via quantile sampling from the empirical distribution, $F_R^{-1}(u)$. The total city-scale emissions for iteration $j$ were then computed as:

$$T_j = \sum_i E_i \times F_R^{-1}(u)$$

where $E_i$ represents the baseline annual emissions of road link $i$, and the sampled correction factor was consistently applied to all links within the same road class. This process was repeated 10,000 times ($j = 1, \dots, 10{,}000$), yielding a full ensemble of possible emission totals. From the resulting Monte Carlo ensemble, 95% confidence interval was calculated.

## 3 Results

### 3.1 Annual emissions

The total on-road $CO_2$ emissions in 2023 among the 20 cities ranged from 0.4 Mt $CO_2$/yr to 7.9 Mt $CO_2$/yr. The top five emitting cities are Berlin (7.9 Mt), Hamburg (6.6 Mt), Cologne (4.1 Mt), Munich (3.5 Mt), and Rotterdam (3.0 Mt). Berlin's $CO_2$ emissions are approximately 20 times higher than those of Lille, the city with the lowest emissions in the dataset (0.4 Mt). On average, the 20 cities emit 2.4 Mt $CO_2$ per year, with a coefficient of variation of 0.82 (Figure 4a). As shown in Figure 5, the linear regression analyses between on-road $CO_2$ emissions and both urban area and population indicate strong positive relationships. Specifically, $CO_2$ emissions increase significantly with larger urban areas and higher population sizes. The regression model yields a high coefficient of determination with an $R^2$ value of 0.98 when emissions are regressed against area, suggesting that urban land extent is a dominant factor influencing total emissions. A similarly positive but weaker correlation is observed between emissions and population, with an $R^2$ value of 0.74, indicating that population size also plays a substantial role in shaping emission levels. This distinction is further illustrated by a comparison between Paris and Hamburg. While their populations are relatively similar, Hamburg covers an urban area nearly seven times larger than that of central Paris. Furthermore, Hamburg's road network is more than three times as long. As a result, Hamburg exhibits substantially higher on-road $CO_2$ emissions, reinforcing the observation that urban spatial extent and infrastructure scale are critical determinants of total emissions, potentially more so than population alone.

Table S8 compares the annual emissions estimated in this study with those reported by Carbon Monitor and other available data sources. Carbon Monitor provides $0.1° \times 0.1°$ daily gridded maps named GRACED (Dou et al., 2023). City boundaries were applied to clip GRACED grids, and area-weighted daily emissions were aggregated to annual city-level totals. Available

data of several cities from Climate Trace (Kott et al., 2024), local statistical websites (Bilanz des Statistikamtes Nord, 2024),
and previous studies (Kühbacher et al., 2023; Ulrich et al., 2023; Anjos and Meier, 2025) was also collected. Overall, estimates
of other datasets are much lower than this study, with differences ranging from −94.2% (Nice, Carbon Monitor) to −8.1%
(Berlin, Ulrich et al.'s estimates from Opendata) relative to our estimates. These discrepancies can be explained by the methods
of different datasets. Compared with local statistical reports, our estimates tend to be higher because we include emissions
from vehicles traveling across city boundaries, whereas local statistics typically estimate emissions based only on oil
consumption within administrative limits. GRACED allocates emissions based on EDGARv5 using OpenStreetMap data
without actual traffic volume data, this method likely underestimates emissions in large cities with high-volume roads. Climate
Trace estimates average annual daily traffic (AADT) by integrating Sentinel-2 satellite imagery with AADT data from the U.S.
Department of Transportation's Federal Highway Administration (FHWA), applying Convolutional Neural Network and
Graph Neural Network models. This U.S.-centric training may limit the models' applicability in the European context. Finally,
although our approach benefits from a more comprehensive road network, the relatively low accuracy on middle and small
roads may contribute to overestimation of traffic volumes in certain areas, as mentioned in Section 2.4.

Per capita emissions show a mean of 2.8 tons/person with a coefficient of variation of 0.4, and the ranking is quite different
from total emissions (Figure 4b). Some of the cities with high total emissions also have high per capita emissions, such as
Cologne (3.8 t/person), Rotterdam (4.6 tons/person) and Frankfurt (3.6 tons/person). Other cities like Berlin (2.1 t/person) and
Paris (0.9 t/person) exhibit low per capita values despite their large total emissions. Notably, cities such as Toulouse (3.4
tons/person) and Marseille (3.2 tons/person) have high per capita emissions, highlighting differences in cities' boundaries e.g.,
including or not satellite towns commuting with each 'city', transportation infrastructure, commuting patterns, and vehicle
efficiency across the regions. Figure 4c illustrates the emissions per unit area, revealing a contrasting pattern to total emissions.
Paris exhibits the highest emissions per unit area (0.02 Mt/km$^2$), despite having one of the lowest per capita values, which is
indicative of its dense urban environment and intensive transportation activities within a compact city layout and a very dense
street network. Similarly, Toulouse ranks second in per-area emissions, despite being only seventh in total emissions. This
result shows that urban density and mobility intensity significantly influence emission distribution at the local scale.

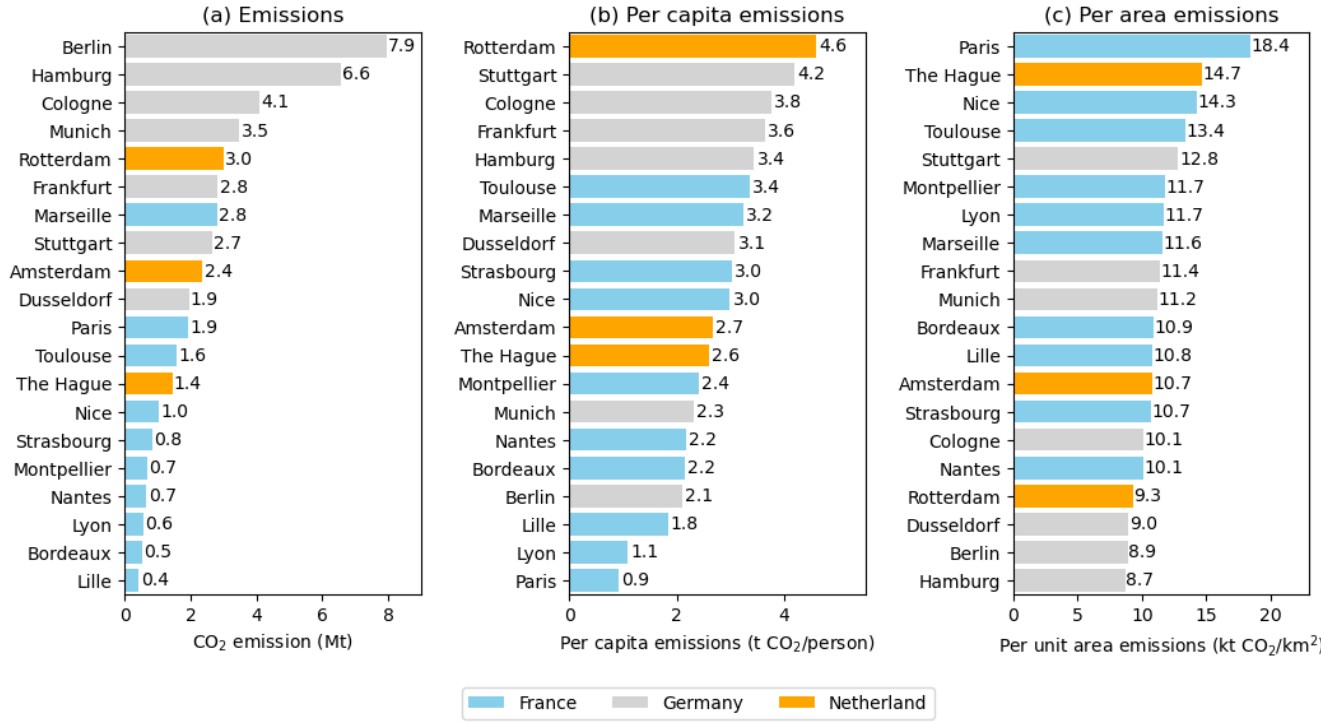


**Figure 4: Annual CO₂ emission and emission intensities per capita and per unit area of 20 cities in 2023.** Grey, light blue and orange represent cities in Germany, France and the Netherlands, respectively.


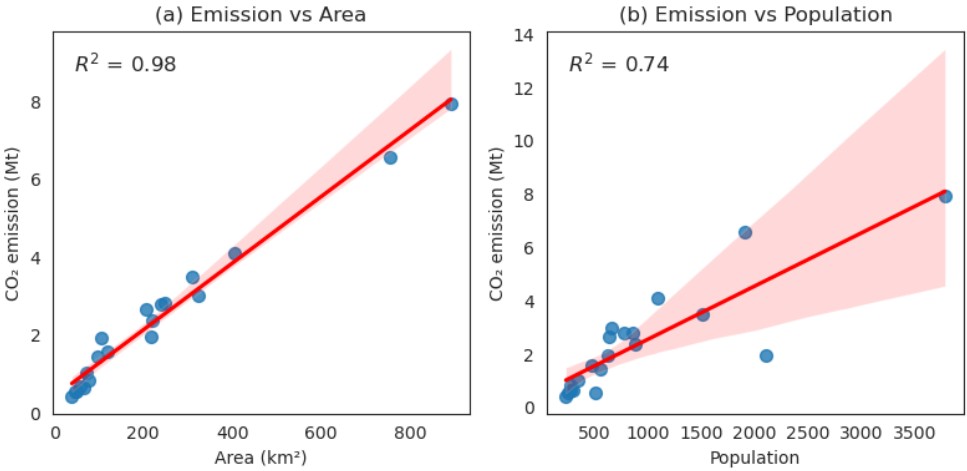

**Figure 5: Linear relationships between on-road CO₂ emissions, area, and population. Each point represents one city.**

## 3.2 Spatial patterns

Figure 6 presents the annual emission maps for 20 major European cities, highlighting the diversity in emission spatial patterns. In addition, two cities from each country were selected to plot cumulative emission curves, as shown in Figure S5. In cities such as Paris, Amsterdam, The Hague and Dusseldorf, a few major roadways stand out significantly in bright yellow. In Paris, the top 5% of the highest-emitting 100 m grids contribute 33.1% of total emissions. The ring road known as le Périphérique emerges as a major hotspot, accounting for 26.9% of the city's total on-road emissions and having a mean emission level that is 953.3% higher than the city-wide average. This is primarily attributable to its high traffic density and heavy vehicle usage driven by significant commuter flows. A similar concentration of emissions is observed in Amsterdam, where the top 5% of the highest-emitting 100 m grids contribute 30.3% of total emissions, respectively, underscoring the spatially skewed distribution of traffic-related $CO_2$. The top 5% of high-emission grids in The Hague and Dusseldorf show a lower contribution of total emissions (24.5% and 21.9% respectively), but these are still concentrated along major highways such as the A4 and A12 in the Hague and B8 and A44 in Dusseldorf. The steep curvatures at the start of the cumulative emissions distribution curves for these two cities suggest that only a few key segments are disproportionately responsible for emissions, albeit to a lesser extent than in Paris or Amsterdam.

Cities like Berlin and Bordeaux exhibit a more diffuse emission pattern, with relatively less pronounced hotspots, where the top 5% of the highest-emitting 100 m grids contribute ~19.0% of total emissions. Their cumulative emission curves demonstrate gentler slopes, indicating a more uniform spread of emissions across the road network. This suggests that no single road or corridor dominates in terms of emission contributions and that urban transport emissions are more evenly distributed. Other cities, including Lyon, Marseille, Frankfurt, and Rotterdam, fall between these two extremes, exhibiting varying degrees of emission concentration. For instance, Frankfurt shows notable linear patterns corresponding to high-emission highways intersecting the urban core. In contrast, Rotterdam reveals both concentrated and dispersed emission zones due to its mixed land use and logistic traffic. Overall, these spatial variations emphasize the importance of city-specific mitigation strategies. While targeted interventions on a few high-emitting corridors may yield significant benefits in cities with highly skewed distributions (e.g., Paris or Dusseldorf), broader, network-wide policies may be necessary in more evenly distributed urban contexts like Berlin or Bordeaux.

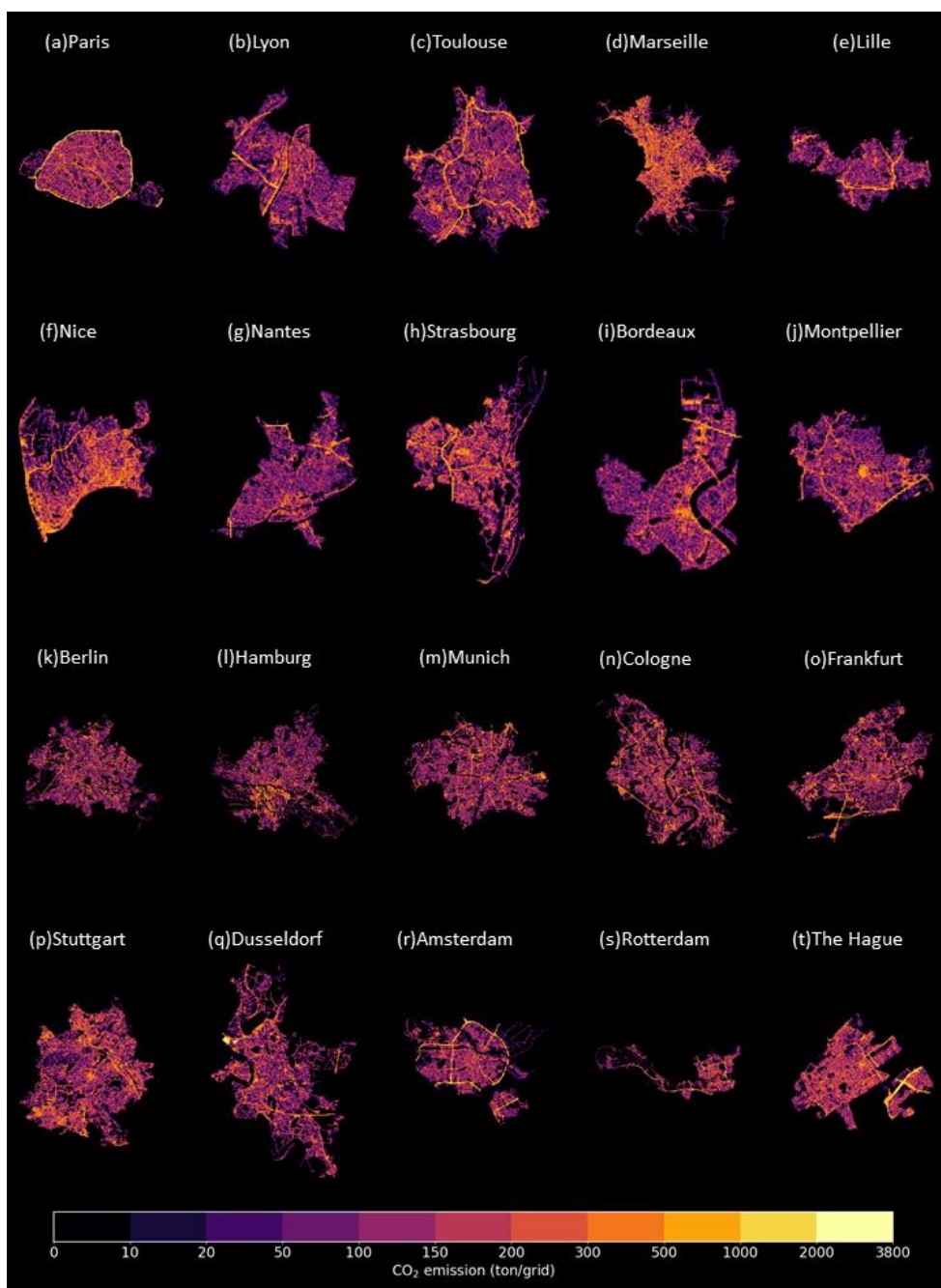

**Figure 6:Annual CO₂ emission map of 20 cities at 100m × 100m resolution in 2023.**

### 3.3 Temporal patterns

Figure 7 presents the normalized daily $CO_2$ emissions ratios for Paris, Berlin, Munich, Amsterdam, Lyon, Marseille, and Nice in 2023. The y-axis represents each day's $CO_2$ emissions divided by the city's total emissions in 2023. These cities were selected due to the availability of corresponding Carbon Monitor Cities data (hereafter CM-Cities data, shown as green dashed lines), which enables direct comparison with the results of this study (blue lines). The time series data reveals distinct seasonal and weekly variations. The summer months (July and August) show a significant decline in emissions in Paris, Amsterdam, and Lyon, while emissions in all seven cities decline around Christmas, due to business closures and decreased commuting. For weekly patterns, there is a slight upward trend from Monday to Friday, a noticeable drop on Saturday, and a further decline on Sunday (Figure S6). The magnitude of the weekend drop varies across cities. In Berlin and Marseille, the median emissions on Sunday drop by approximately 31.1% and 27.7% compared to Friday in 2023, respectively, representing the most pronounced Sunday reduction among the six cities. In contrast, Amsterdam exhibits a much smaller Sunday drop compared to Friday (10.1%).

In all cities, the median emissions of public holidays (marked in grey shades) and school holidays (marked in light blue shades) are lower than those of weekdays in 2023. Across all six cities, the median emissions on public holidays and school holidays were consistently lower than weekday levels in 2023, indicating a general reduction in traffic-related $CO_2$ emissions during holiday periods. In Paris, public holiday emissions were exceptionally low, even lower than Sunday levels by 5.2%. The pattern is different in Marseille, Berlin, and Nice, as the median emissions on public holidays exceeded those on Saturdays by 24.4%, 11.0%, and 6.4%, respectively. The medians of school holidays are generally higher than those of public holidays because a more limited segment of the population is affected, and the distributions are notably wider. An exception is Amsterdam, where public holiday emissions slightly surpassed those during school holidays, suggesting a different urban rhythm or school break dynamics compared to other cities. Also, the day of the week on which a holiday falls also influences emission levels. As shown in Figure S7, holidays that coincide with weekends tend to show similar emission levels to regular weekend days. When holidays fall on a Monday, their emission levels are comparable to those of regular Mondays in cities like Berlin, Marseille, and Nice.

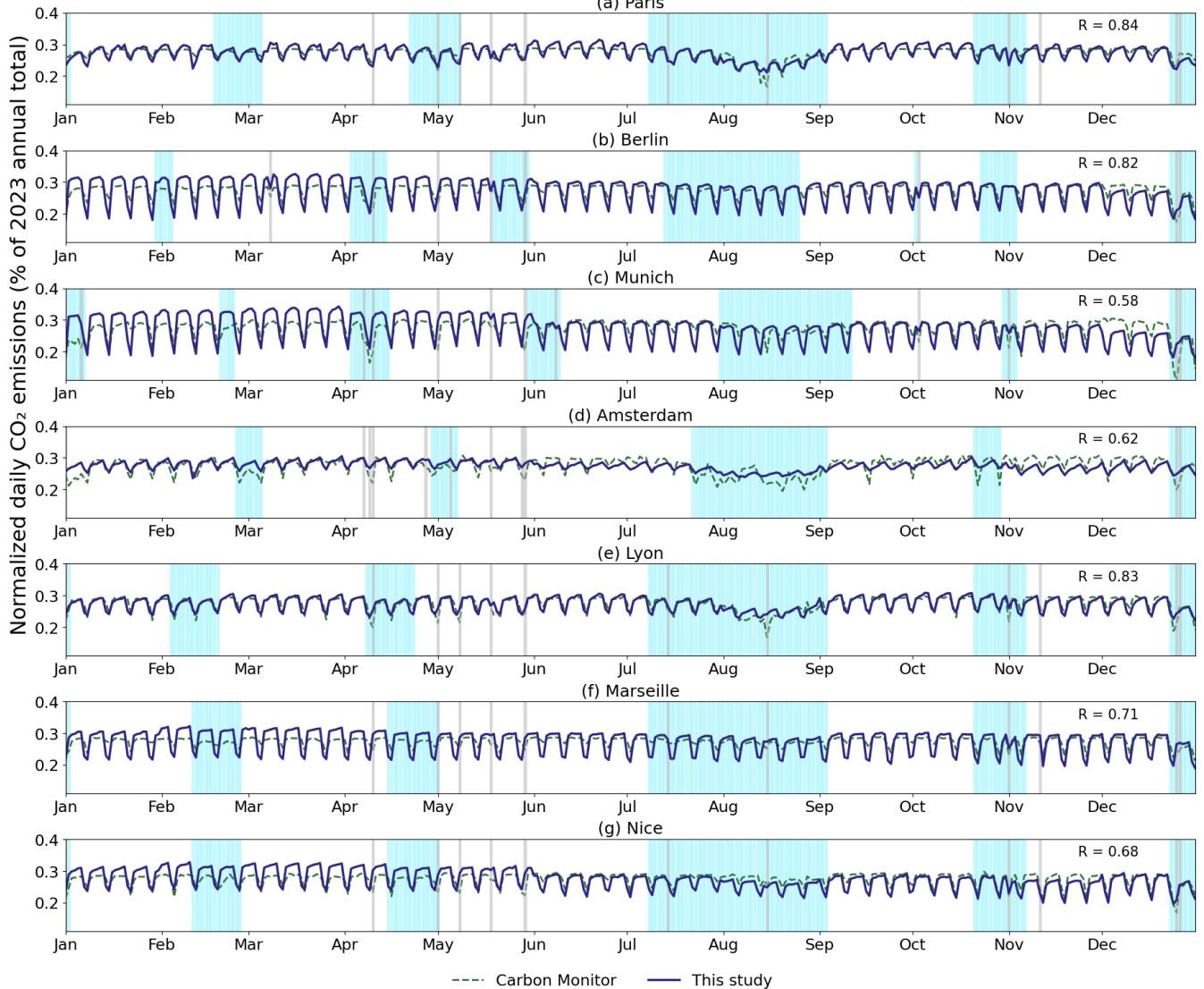

420

**Figure 7: Normalized daily CO₂ emission of seven cities in 2023.** Y-axis represents each day's CO₂ emissions divided by the city's total emissions in 2023. The light blue and grey shades represent school holidays and public holidays, respectively. The y-axis represents each day's CO₂ emissions divided by the city's total emissions in 2023.

Although the general emission temporal variability estimated in this study align reasonably with those reported by Carbon Monitor Cities, as evidenced by the R correlation coefficients ranging from 0.58 to 0.84 across the six selected cities, notable differences remain. In Paris, CM-cities tends to underestimate both the troughs and peaks of emissions (Huo et al., 2022). In Lyon, the consistency is relatively high, but the sharp weekend emission drops observed in Carbon Monitor estimates are not reproduced in this study. In Amsterdam, this study does not show the pronounced weekend decreases during holidays that are

present in Carbon Monitor data. CM-cities estimated traffic volumes using a sigmoid regression based on TomTom live congestion indices, which lack spatial granularity (only one value per city), and the model parameters were calibrated using real-time data from approximately 60 roads in Paris. In addition, CM-cities adopts the Functional Urban Area (FUA) definition used by the OECD and the European Union, which includes high-density urban centers along with their surrounding commuting zones, whereas our analysis relies on administrative boundaries. For cities not covered by CM-cities, we compared daily emissions clipped from GRACED (Figure S8). Without calibration at the city level as CM-cities did, GRACED daily emissions fail to show a consistent weekday–weekend pattern, and some anomalous peaks occurred (e.g., elevated emissions in Hamburg in April 2023 and in Frankfurt and Montpellier in late May 2023). Except for The Hague, Rotterdam, and Bordeaux, the resulting daily profiles showed very poor agreement (R<0.4). These findings suggest that coarse-resolution data are not suitable for city-level temporal analyses, highlighting the advantage of our city-scale dataset in more accurately representing actual urban emissions.

Figure 8 presents the average hourly $CO_2$ emission patterns for cars across the 20 cities in 2023. The y-axis represents the average proportion of daily $CO_2$ emissions for each hour, categorized by day types: holidays (blue), weekdays (orange), and weekends (green). The hourly patterns for cars in French cities and Dutch cities are similar. On weekdays, there are two emission peaks at 9:00~10:00 and 18:00~19:00 due to commuting, and the emissions stabilize at relatively high levels between these two peaks. After the second emission peak, the emissions decline continuously and reach their lowest point at 4:00 ~ 5:00. The differences between weekdays and holidays are relatively small, but with no or a less pronounced morning peak due to reduced commuting activity. On weekends, the sum of average emission share in French cities and Dutch cities during evening and early morning (22:00 to 6:00) reach 22.9% to 29.1%%, significantly higher than that for weekdays (17.4 to 21.7%), and the first peak is lagged to around 12:00. German cities on weekdays, except for Dusseldorf, the $CO_2$ emission exhibit earlier morning peaks at 8:00 and a much higher peak around 15:00 ~16:00. On average, evening peak emissions in French and Dutch cities are only around 15% higher than morning peak levels, but for German cities specifically, the difference ranges from 9.3% to 60.0%. After the peak, the $CO_2$ emissions in German cities decrease sharply, which is consistent with the trends reported by the Berlin datasets (Max et al). On weekends, there is only one peak around 13:00.

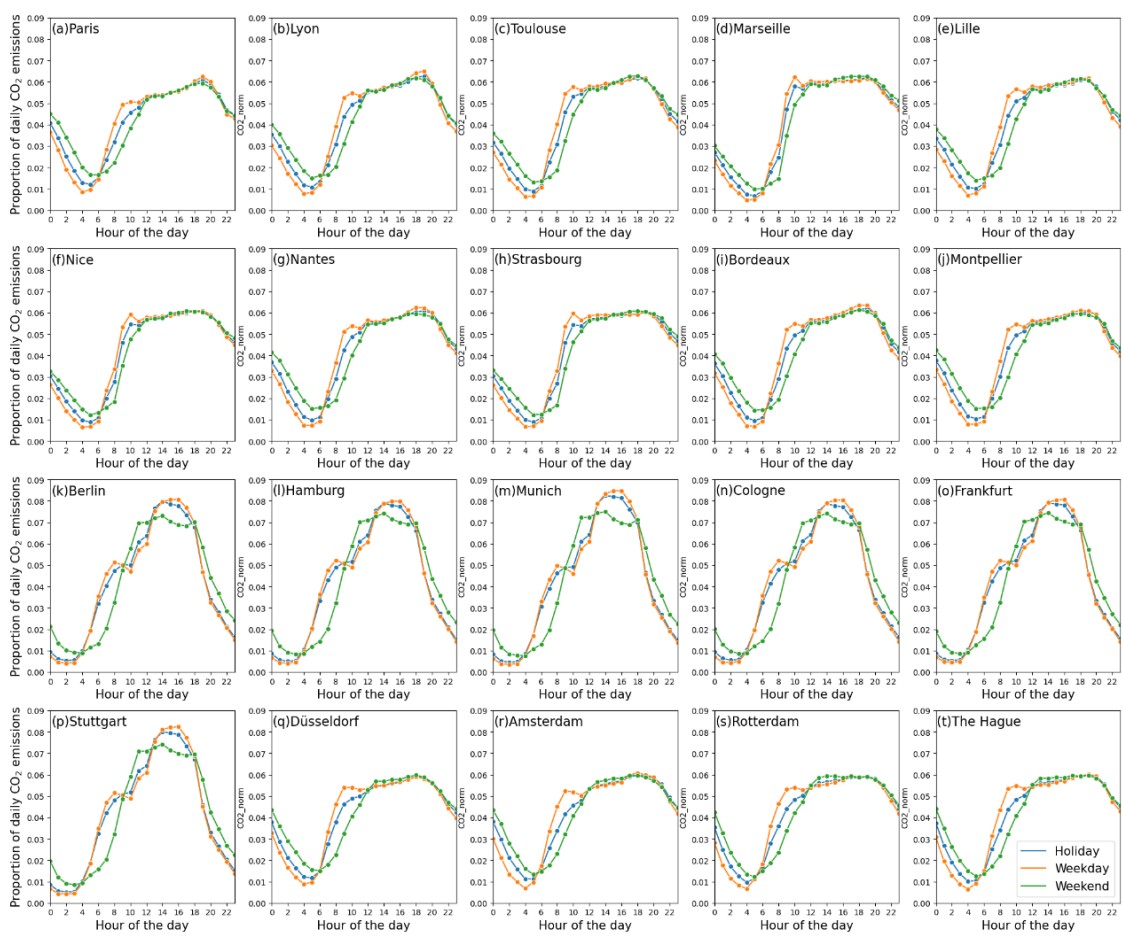

**Figure 8: Hourly emission patterns of cars in 20 cities.**

The hourly patterns for trucks are relatively consistent across all 20 European cities but are notably different from those of
passenger cars (Figure 9). On weekdays, truck-related $CO_2$ emissions show a peak around 9:00 in nearly all cities, suggesting
synchronized delivery and logistics activity. This peak accounts for 5.4%–6.5% of daily truck emissions in French and Dutch
cities, and up to 9% in German cities such as Berlin and Hamburg. Truck emissions on weekends and holidays are considerably
reduced, with no discernible peaks in most cities. In some German cities (e.g., Stuttgart and Dusseldorf), truck emissions
remain below 3% of daily total at any hour during holidays, reflecting stricter weekend freight regulations. In contrast,
emissions levels of trucks remain relatively high on weekends, especially in southern cities like Marseille and Nice, where
midday peaks surpass 0.06 of daily emissions and are comparable to weekday levels.

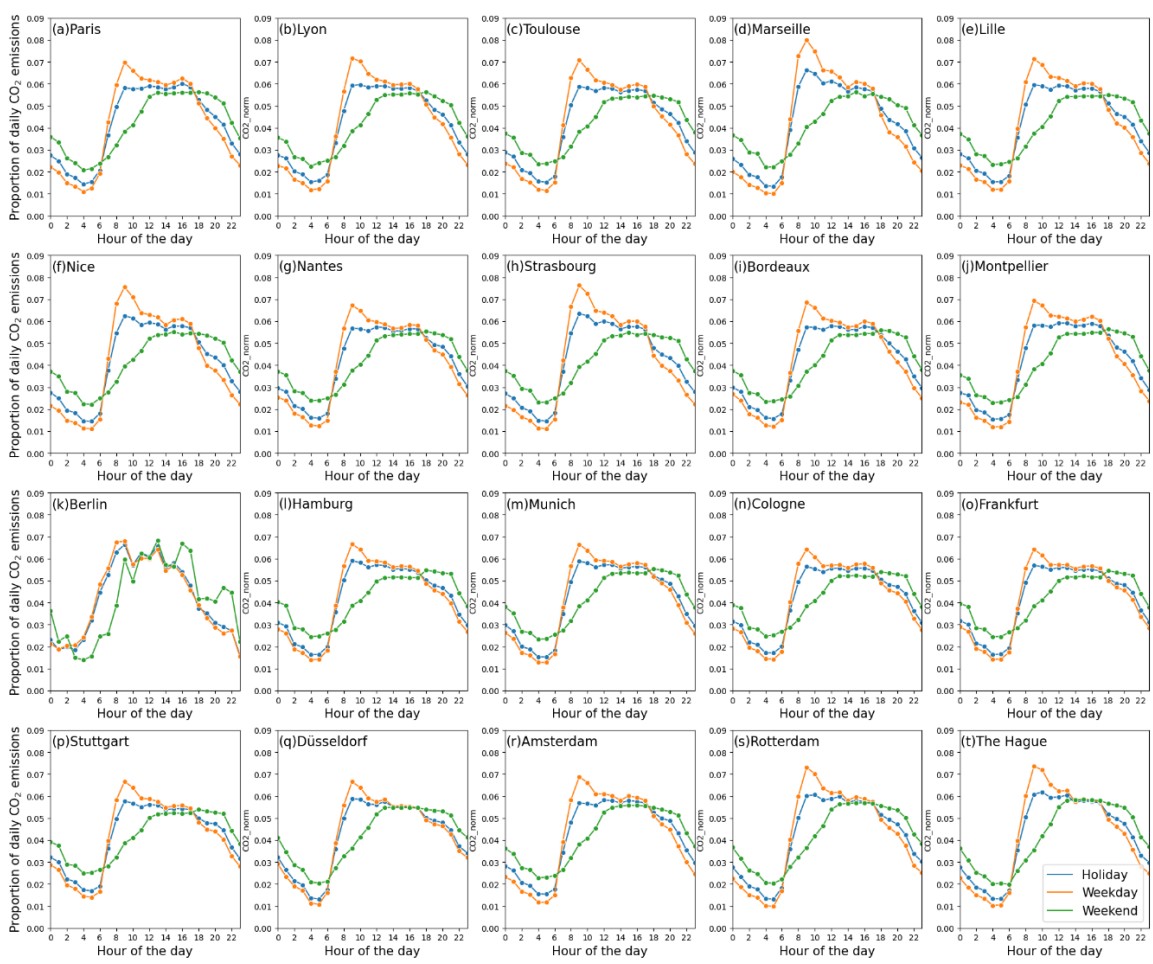


**Figure 9: Hourly emission patterns of trucks in 20 cities.**

**3.4 Uncertainty analysis**
Figure 10 shows the uncertainties in annual emissions arising from uncertainty in traffic volume estimates. Overall, the Monte
Carlo–derived mean emission estimates are close to the original deterministic estimates for most cities, with the Monte Carlo
means being on average 13.1% lower across the 20 cities. the differences between the Monte Carlo mean and the deterministic
estimate for Paris (−7.0%), Lyon (+7.3%), and Bordeaux (−13.4%) remain within ±15%, indicating relatively stable estimates
despite uncertainty propagation. Noticeable differences are observed for Berlin (−41.4%), Hamburg (+61.4%), Marseille
(−41.2%), and Toulouse (−46.5%), where the differences between the Monte Carlo mean and the deterministic estimate exceed

475 40%.


Figure S9 further shows the road-class-specific uncertainties. Across cities uncertainty in annual totals is primarily driven by
emissions associated with small roads, which exhibit the greatest relative variability across all functional classes. We quantify
road-class-specific relative uncertainty using the relative 95% interval width defined as (P97.5−P2.5)/mean of the 10,000
Monte Carlo realizations. Using this metric, small roads show the largest relative uncertainty, with a median value of 2.67
(266.7%), compared with 1.74 (174.1%) for middle roads and 1.26 (125.8%) for major roads. In Berlin, the Monte Carlo
estimate is 4.65 Mt $CO_2$ (95% CI: [1.89,6.04]) , closer to values reported by Anjos et al (2.70 Mt)(Anjos and Meier, 2025) and
Climate Trace(1.99 Mt)(Kott et al., 2024), suggesting that the original deterministic estimate may have overestimated
emissions from small roads. The situation in Hamburg is different. The Monte Carlo mean emission estimate of approximately
10.57 Mt $CO_2$ (95% CI: [5.64, 15.60]) exceeds that of Berlin, which is unreasonable given Hamburg's smaller urban scale and
lower overall road lengths. This outcome suggests that limited and heterogeneous observational data can bias an upward bias
in the sampled correction factors for small roads, resulting in an overestimation of emissions for this road class and,
consequently, at the city scale.

Overall, these contrasting behaviours highlight that city-scale uncertainty is highly sensitive to the treatment of small roads,
particularly in data-scarce contexts. While the Monte Carlo framework provides a robust characterization of uncertainty, its
outcomes for low-traffic road classes should be interpreted with caution and ideally complemented by additional constraints
or external benchmarks.

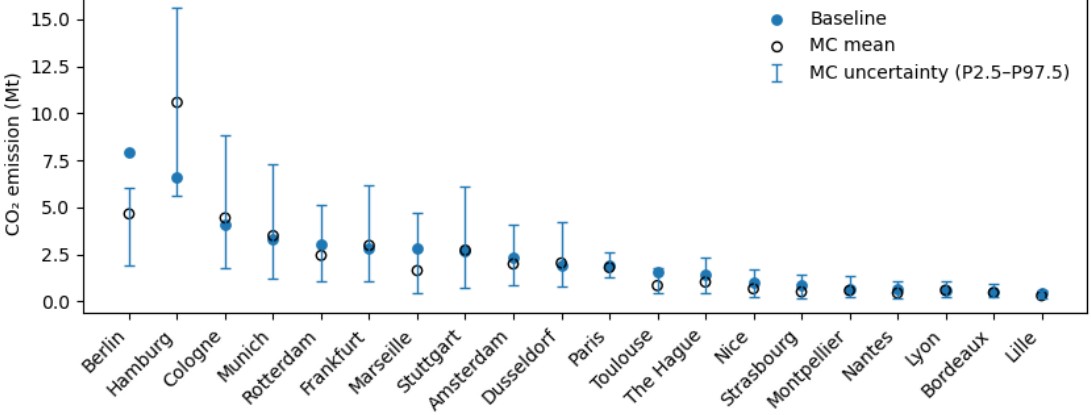

**Figure 10: Emission uncertainties in 20 cities. Filled circles are the original deterministic estimates. Hollow circles indicate Monte**
**Carlo mean estimates, and vertical bars represent the 95% uncertainty interval (P2.5–P97.5).**

**4 Discussion**

**4.1 Key contributions and implications**

This study demonstrates that integrating new GPS-based traffic data for individual vehicles covering all street segments with the COPERT model enables the estimation of hourly on-road $CO_2$ emissions at street level, which were further aggregated into $100 \times 100$ m grids for visualization, to generate high-resolution emission maps across 20 European cities. This approach overcomes the limitations of traditional top-down downscaling methods (e.g., population-based or road-network density proxies) by applying machine learning to impute the actual traffic volumes from FCD, which only samples the traffic of vehicles equipped with GPS. Compared to existing $CO_2$ emission inventories such as CAMS-TEMPO, Carbon Monitor, or localized platforms, our dataset represents a significant advancement by simultaneously achieving high spatial granularity and temporal resolution. It captures intra-urban variability that is often missed in coarser-resolution datasets or those relying solely on major road segments. This work highlights the value of integrating GPS-based mobility data with machine learning and emission modelling to enhance the monitoring of urban transportation emissions and to inform the design of effective, location-specific mitigation policies. Most common low-carbon transport measures in cities include modal shift to public transport, low-carbon zones control, and low-emission vehicle development, but each strategy may vary according to development stages and types of urban land-use transport systems (Creutzig et al., 2012; Nakamura and Hayashi, 2013; Croci et al., 2021). While low-density cities become more compact in the long term but often lack sufficient population density to support rapid transit systems in the short term, promoting the adoption of electric vehicles, particularly in regions with low-carbon electricity, may be a more practical approach (Kennedy et al., 2014). This study may support the design of such strategies by enabling street-level scenarios to quantitatively assess their potential emission reductions.

Our hourly $CO_2$ emission maps reveal striking spatial heterogeneity within cities. For example, concentrated emission hotspots along Paris' ring road, versus more dispersed patterns in Berlin, reflect differences in urban structure, transport systems, and commuting behaviours. Temporally, we observed national variations in traffic-related emissions during holiday and summer periods, likely due to country-specific vacation schedules. Our new emission maps can support planning of low-emission zones, help identify high-flux corridors for targeted energy efficiency measures and provide a basis for congestion-related studies. Given that traffic congestion is a major driver of both fuel consumption and emissions, our maps offer valuable insights for designing and evaluating emission reduction strategies.

**4.2 Limitations**

Several sources of uncertainty remain in our approach. Because the GPS-to-volume conversion models were calibrated using in-situ sensor data from Paris and Berlin only and extrapolated to the remaining 18 cities, the results may be better suited for analysing spatial patterns, temporal dynamics, and relative differences across cities, rather than for precise reporting of absolute

emission magnitudes. To move beyond qualitative statements, we quantify activity-data uncertainty using independent annual AADT/AAWT validation (Section 2.4; Figure 3) and Monte Carlo uncertainty propagation (Section 3.4; Figure 10 and Figure S9). The external validation reveals pronounced inter-city heterogeneity in traffic-volume agreement (with $R^2$ ranging from approximately 0.3 to 0.92 across cities; Figure 3), which provides the empirical basis for the subsequent uncertainty ranges.

First, significant uncertainty may be introduced during the conversion from GPS trajectories to actual traffic volume. The flux-to-volume machine learning models were calibrated using sensor data from Paris and Berlin only, because comparable high-resolution traffic counts are either unavailable or not publicly accessible for most other cities. In addition, GPS penetration rates may vary across cities and vehicle types, and the vehicle population captured by FCD may differ from that represented in local monitoring stations, which can affect calibration, particularly for trucks. As discussed in Sections 2.4 and 3.4, model performance is weaker on middle and small roads, and emissions from small roads exhibit the largest uncertainty and potential overestimation. Consistent with this, Monte Carlo mean emission estimates are on average 13.1% lower than the deterministic totals across the 20 cities, and most cities remain within ±15%. However, several cities show substantially larger deviations exceeding 40% (e.g., Berlin, Hamburg, Marseille, and Toulouse), indicating that absolute totals are more uncertain where traffic-volume discrepancies are large and observational constraints are limited. For example, Berlin's Monte Carlo estimate is 4.65 Mt $CO_2$ (95% CI: [1.89, 6.04]), whereas Hamburg shows a much wider and higher range of 10.57 Mt $CO_2$ (95% CI: [5.64, 15.60]), highlighting the sensitivity of city totals to correction factors on small roads in data-scarce contexts. This reinforces the need for more comprehensive and standardized traffic monitoring networks. Incorporating additional top-down constraints, such as city-level fuel consumption statistics in transportation sector, could further improve the accuracy of traffic volume inference.

Second, uncertainties also arise from fleet structures. Due to the lack of detailed vehicle-type distribution at the road segment level, we can only perform fleet correction for roads where heavy-duty vehicle traffic is explicitly restricted. For other roads, we currently apply city-wide average fleet compositions, which may not reflect local variations. Although urban fleet structure evolves continuously, available data are reported at coarse temporal resolution; disaggregation to finer temporal scales would introduce substantial uncertainty, and an annual fleet update is therefore adopted to maintain consistency with the data and the emission modelling framework.

Finally, emissions in this study are estimated using the COPERT, which is based on an average-speed framework and does not explicitly represent microscopic stop-and-go driving behaviours. In contrast, microscopic emission models such as MOVES(USEPA, 2024) explicitly account for such dynamics but require high-frequency trajectory data, which are not available in this study. Moreover, COPERT characterizes vehicle technologies primarily by vehicle category and Euro emission standard, and does not explicitly parameterize changes in emission performance associated with vehicle ageing. As a result, city-specific fleet age structures and local real-world driving conditions may lead to deviations from the standard

emission factors used in the model, especially where detailed fleet data are unavailable to further refine the parameterization. Access to locally measured emission factors from in situ studies or the literature would help reduce this source of uncertainty and improve the accuracy of the emission estimates.

**4.3 Future work**

Current work only covers the year 2023, but the underlying GPS-based FCD is typically available with a delay of only about one week. This creates a clear opportunity to automate the processing pipeline and update the emission estimates on a rolling basis. Incorporating this capability into Carbon Monitor Cities would allow near-real-time, high-resolution emission monitoring at the street level, significantly enhancing the system's responsiveness and value for both research and policy applications. In addition, further feature engineering could improve model performance. As part of ongoing work, we plan to incorporate high-resolution urban context information, such as building-type data, to better capture heterogeneity across different road classes. The proposed framework is flexible and allows additional features to be integrated as new data become available. Also, future work could extend the methodology to include major air pollutants beyond $CO_2$ and scale the approach to cover broader regions. Through incorporating more sensor-based traffic measurements across cities, data representativeness and model validation can be further improved. Such efforts will strengthen the robustness, applicability, and policy relevance of street-level emission mapping, particularly in supporting timely decision-making and climate or clean air action monitoring.

**5 Data availability**

The high-resolution hourly $CO_2$ emission dataset for 20 cities in 2023 is available in NetCDF format, on Zenodo **https://doi.org/10.5281/zenodo.16600210 (Shi et al., 2025)**. Each city has an individual NetCDF file that provides gridded hourly emissions over the entire year of 2023. Their central x and y coordinates define the grid cells, and each file includes the variable CO2_g, representing emissions in grams per hour in the grid. Every grid's size is 100 m × 100 m.

**Supplement**

This dataset is accompanied by Supplementary Information, including a detailed methodology document (SI_document.docx) and additional data tables (SI_tables.xlsx).

**Author contributions**

QS processed and generated the dataset and drafted the initial manuscript. PC designed the study and provided scientific supervision. NM collected the raw data and contributed to the structuring of the dataset. XB and RTM assisted with COPERT data handling, data matching, and emission calculations. RE reviewed the emission estimates and provided constructive

feedback on the manuscript. CZ contributed extensively to the machine learning modelling and provided valuable suggestions on the manuscript structure and visualization. All authors reviewed and approved the final manuscript.

## Competing interests

The authors declare that they have no conflict of interest.

## Acknowledgement

This study is funded by the Copernicus Atmosphere Monitoring Service (under the CAMS2_51a contract), which is implemented by the European Centre for Medium-Range Weather Forecasts (ECMWF) on behalf of the European Commission.

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
