# Peer review of "High spatiotemporal resolution traffic CO2 emission maps derived"

_Earth System Science Data, 2025_

## Author Response (AR1)

**#Reviewer 1**

The paper clearly presents the resource described in the title.

Probably the authors have already planned this, but updating the fleet data at regular intervals is crucial for model accuracy given the fairly rapid pace of vehicle electrification in Europe.

I was curious about fuel usage with start/stop driving, e.g. in heavy freeway traffic and urban cores. The fuel consumption for start/stop can be very different than the hourly mean assuming a constant pace over a given distance.

I suggest the authors add a table in the manuscript listing the vehicle types considered. For example, in some cities policies differentiate light truck and delivery traffic from passenger traffic. And one could imagine policies to promote mopeds. But without reading the SI, as a reader I don't know whether your data can differentiate these vehicle types.

**Author's response:**

We thank the reviewer for this insightful comment and fully agree that fleet composition and driving dynamics ideally should be updated at the highest possible temporal resolution.

In this study, fleet data are compiled from publicly available municipal and statistical sources, which are typically updated annually. Attempting to extrapolate annual fleet statistics to a higher temporal resolution would introduce substantial uncertainty. We therefore update fleet composition annually, which ensures consistency with the available data and the overall modeling framework. We added the discussion in L552 – L555: Although urban fleet structure evolves continuously, available data are reported at coarse temporal resolution; disaggregation to finer temporal scales would introduce substantial uncertainty, and an annual fleet update is therefore adopted to maintain consistency with the data and the emission modelling framework.

Regarding start–stop driving behavior, we fully agree that it can substantially affect real-world fuel consumption and emissions under congested conditions. However, emissions in this study are estimated using the COPERT model, which is based on an average-speed framework and does not explicitly resolve microscopic stop-and-go dynamics. Vehicle models such as MOVES can realize that but require high-frequency trajectory data, which are not available here. We added the clarification of this limitation in the revised manuscript and note that more detailed data and microscopic models would be required for a robust quantification (L557 – L560):

Finally, emissions in this study are estimated using the COPERT, which is based on an average-speed framework and does not explicitly represent microscopic stop-and-go driving behaviours. In contrast, microscopic emission models such as MOVES(USEPA, 2024) explicitly account for such dynamics but require high-frequency trajectory data, which are not available in this study.

Also as suggested, the vehicle types considered are added in Table 4.

**Table 4: Vehicle categories**

| Big Category | Category | Fuel |
|---|---|---|
| Car | L-Category | Petrol, Diesel |
| | Buses | Petrol, Diesel, CNG, Diesel Hybrid, Biodiesel, Battery electric, Diesel PHEV |
| | Passenger Cars | Petrol, Diesel, CNG, Petrol Hybrid, Petrol PHEV, Battery electric, Diesel PHEV |
| Truck | Heavy Duty Trucks | Petrol, Diesel, Diesel PHEV, Battery electric, CNG |
| | Light Commercial Vehicles | Diesel, Petrol, Diesel PHEV, Battery electric, CNG, Petrol Hybrid, Petrol PHEV |

**#Reviewer 2**

This work developed a new approach for hourly CO2 emission mapping at high resolution from on-road traffics for 20 cities in France, Germany, and the Netherlands in 2023, with the FCD data created from GPS information, the traffic volume data based on machine leaning models, and speed- and vehicle type-specific emission factors. The new CO2 emissions from on-road transportation were validated and the spatial and temporal variation characteristics were presented and discussed. The manuscript is generally well written. There are some comments which are required to be addressed before it can be accepted.

**Author's response:** We thank the reviewer for the detailed comments. Point-by-point response is followed, and the added text are highlighted in the annotated PDF provided as an attachment.

1.    Pay attention to the blank before the bracket particularly in the Introduction Section.
Revised.

2.    Line 13, point out the $CO_2$ emission from on-road traffic or transportation.
Revised.
L12: In this study, we developed new hourly on-road $CO_2$ emission maps with a 100 × 100 m resolution for 20 major cities in France, Germany, and the Netherlands in 2023.

3.    Line 44-55, the details of this paragraph are not so necessary. Simplify the sentences and link them more to the major contents of this study.

This paragraph aims to describe the carbon emission reduction actions in cities, then propose the importance of high-resolution emission maps in the next paragraph. As suggested, we simplified it.

L44 – L52: Emission reduction targets are being translated into concrete actions at the city level. For instance, the transport sector is responsible for approximately 20% of Paris' local greenhouse gas emissions (Albarus et al., 2025), and Paris plans to reduce its direct emissions by 50% by 2030 and 100% by 2050, compared to 2004. Paris has set itself the target of phasing out diesel-powered mobility by 2024 and petrol-powered mobility by 2030, aligning with the EU-wide ban on the sale of internal combustion engine vehicles by 2035. Amsterdam aims to achieve zero-emission transport by 2030, phasing out all fossil-fuel vehicles within city limits (Amsterdam, 2024). The city is rapidly expanding its electric vehicle infrastructure, as all newly registered vehicles are required to have zero-emission engines in 2025 (CINEA, 2025). Similarly, to achieve climate neutrality in 2050, Berlin will require a long-term reduction in $CO_2$ emissions in the transport sector to around 1.17 million tonnes of $CO_2$ per year, a reduction of around 77 % compared with 1990 emissions (diBEK, 2025).

4.    Line 121, The title of Figure 1 is not correct.
Revised as: Figure 1: Workflow of this study.

5.    Line 165, in Table 2, for the road-specific traffic count data, are they daily or hourly? Only hourly traffic volume can be used to produce hourly emissions.

The road-specific traffic count data is hourly. Revised as "Daily mean derived from hourly averages" to make it clearer.

6. Line 195-196, why use monthly average instead of hourly or daily average meteorological data to calculate the emission factors?

This is because COPERT only supports monthly meteorological data as input.

7. Line 206-207, clarify the potential uncertainty caused by using a standard $EF_{CO2}$, instead of a measured $EF_{CO2}$ from literature.

We extend the discussion part.

L560 – L565 : Moreover, COPERT characterizes vehicle technologies primarily by vehicle category and Euro emission standard and does not explicitly parameterize changes in emission performance associated with vehicle ageing. As a result, city-specific fleet age structures and local real-world driving conditions may lead to deviations from the standard emission factors used in the model, especially where detailed fleet data are unavailable to further refine the parameterization. Access to locally measured emission factors from in situ studies or the literature would help reduce this source of uncertainty and improve the accuracy of the emission estimates.

8. Line 345-346, this sentence is repeated.

Deleted L345 – 346.

9. Line 389-390, Is there any difference in the emission factors used for calculations which could cause the discrepancies?

Differences in emission factors across models can indeed contribute to discrepancies, especially when external inventories rely on different EF frameworks or updated calibrations. In this study we apply a consistent COPERT-based EF set across all cities, so the discrepancies highlighted here mainly stem from uncertainties in traffic activity (GPS-derived volumes) rather than EF differences within our calculations. We discuss the activity data uncertainty in more detail in our response to Reviewer 3.

**#Reviewer 3**

This paper is interesting and necessary for quantifying urban carbon emissions. I'm impressed by the authors' effort to present a high-resolution approach for road traffic $CO_2$ emissions. This kind of information is very important for urban planning, both now and in the future. In fact, this work tackles a necessary issue in urban emission monitoring: representing the emissions of individual roads across an entire city scale. The study has many strengths: the claim of providing the first hourly, street-level emissions for 20 European cities is a major achievement, even if the "first time" assertion might be too strong (other studies have done similar work, though not with this many cities).

Furthermore, the methodology is well-structured and transferable, with solid documentation of the data processing steps. The applied use of ML and the choice of the algorithm are, in my view, well-justified. Finally, it covers a considerable spatial extent, including small roads ( e.g., residential classe) that are often omitted from many inventories and even high-definition urban traffic emission studies.

Although I acknowledge the study's importance, several methodological concerns and uncertainties must be addressed by the authors before publication. My review is divided into the following points.

**Author's response:** We thank the reviewer for the detailed comments. Point-by-point response is followed.

1 _____

The most critical weakness of this study based on the extrapolation/generalization of the GPS-to-volume conversion and the resulting $CO_2$ emissions, since this decision uses a ML model trained exclusively on data from Paris and Berlin and applies it to the remaining 18 cities. While the authors justify this by stating that "high-quality in-situ traffic observations are either unavailable or not publicly accessible for other cities," this extrapolation represents a major source of uncertainty that needs better justification and thorough discussion.

The main issue is the assumption that factors, such as GPS penetration rates, fleet compositions, and traffic behavior, in cities like Munich, Amsterdam, and Lyon will match those in Paris or Berlin. This assumption may hold true for some roads but is unlikely to be valid across the board. This simplified idea has strong impacts on the results and their applicability in urban contexts.

Have look at this: the ML model validation showed poor performance on middle and small roads in Paris (car R2 0.33, truck 0.23) and major in Berlin (car 0.66, no truck). I suppose, applying this model to other cities with potentially different GPS penetration rates and urban structures introduces unquantified errors, particularly on those less-traveled routes. Importantly, no sensitivity analysis is provided by authors to quantify how variations in GPS penetration (and, consequently, traffic volume estimates) affect the final emission estimates. To do so, I recommend to perform a sensitivity analysis demonstrating how hypothetical variations in GPS penetration (e.g., ±10%, ±20%, ±30% ) affect the total estimated emissions for the extrapolated cities; provide explicit uncertainty bounds for each extrapolated city, reflecting the potential error introduced by model transferability. This also leads me to second point.

2___

I am not convinced that validation strategy, limited to 2 cities- Paris and Berlin, is sufficient for main claim made about all 20 cities. Note that no independent validation for ~90% of the cities analysed.

I am not sure it if is possible, but I would suggest an external validation to compare the model's volume estimates with any available traffic statistics (even if only annual or from limited sites) from the 18 extrapolated cities.

We thank the reviewer for this detailed and constructive comment. The external validation is conducted at the annual scale by comparing our estimated traffic volumes with independent, publicly available traffic count data from a newly developed dataset (Bonnemaizon et al., 2025). Based on the discrepancies observed in this annual-scale comparison, we then perform a Monte Carlo uncertainty analysis to quantify how uncertainties in traffic volume estimates propagate into the final $CO_2$ emission estimates. We therefore address Comments 1 and 2 together below, as the external validation directly underpins the uncertainty analysis related to model transferability.

We recognized the importance of validating the transferability of the GPS-to-volume conversion model. While, unfortunately, hourly or daily traffic sensor data remain unavailable for cities beyond Paris and Berlin, our colleagues have made a concerted effort to address this limitation by compiling an independent dataset (Bonnemaizon et al., 2025). This dataset was developed by collecting and harmonizing available Annual Average Daily Traffic (AADT, in number of vehicles per day) and Annual Average Weekday Traffic (AAWT, equivalent to AADT excluding weekends) data for European cities. Besides Paris and Berlin, we also obtained another 6 cities's AAWT/AADT in 2023: Montpellier and Hamburg (AADT), and Bordeaux, Lyon, Toulouse and Lille (AAWT). We then use this newly developed dataset to validate our traffic volume estimates, and the results of this comparison are now included in **Section 2.4(Figure 3)**. This additional validation provides further evidence on the robustness of the extrapolated results and informs the uncertainty assessment presented in the revised manuscript.

The result was added in **Section 2.4, L247 – L261**: In addition to Paris and Berlin that are used for model training, observed traffic-count-based annual average daily traffic flow (AADT, in number of vehicles per day) or annual average weekday traffic (AAWT, equivalent to AADT excluding weekends) datasets are available for six additional cities reported in a recent study(Bonnemaizon et al., 2025): Montpellier and Hamburg (AADT), and Bordeaux, Lyon, Toulouse and Lille (AAWT). The comparison which serves as independent external validation to assess our traffic volume estimates is shown in Figure 3. Paris, the most important reference city for model development, shows strong agreement between estimated and public AADT values ($R^2$ = 0.92, n = 2696), with data points across all road classes closely aligned with the 1:1 line. Berlin exhibits noticeably larger dispersion, with a moderate $R^2$ (0.55) derived from a relatively small sample size (n = 197), which likely contributes to the lower correlation.

Lyon, Hamburg, Bordeaux and Montpellier all show moderate correlation (with $R^2$ around 0.6). However, while simulated and observed traffic volumes are generally well aligned for Bordeaux, public observations for Lyon, Hamburg and Montpellier tend to exceed the simulated values,

especially for the major roads. Toulouse and Lille are characterized by low correlation (R² around 0.3), exhibits the weakest consistency between estimated and public traffic volumes. Overall, the scatter plots reveal pronounced city-to-city heterogeneity in traffic volume agreement, providing important context for subsequent uncertainty propagation to city-scale emission estimates.

[Figure]

**Figure 3: Comparison of AADT/AAWT between this study and public datasets**

Monte Carlo method is widely used in emission studies to estimate uncertainties. Rather than relying on hypothetical perturbations of GPS penetration rates (e.g., ±10%, ±20%, ±30%) as initially suggested, we adopted a Monte Carlo approach that propagates empirically observed discrepancies between estimated traffic volumes and independent public AADT/AAWT datasets into city-scale emission estimates.

Since $CO_2$ emissions scale linearly with traffic volume, uncertainty in traffic counts can be directly translated into uncertainty in emissions. Instead of assuming a predefined distribution (e.g., normal or lognormal), we adopt a non-parametric approach using empirical cumulative distribution functions (ECDFs) derived from the observed discrepancy ratios. These ratios are grouped by road class (major, middle, small), and for each Monte Carlo iteration, random correction factors are sampled from the ECDF and applied to all road links of the corresponding class. This process is repeated 10,000 times, generating an ensemble of possible emission totals for each city. From this ensemble, we derive the 95% confidence intervals that reflect the propagated uncertainty due to volume estimation errors. This method allows for data-driven uncertainty quantification without relying on strong parametric assumptions, and directly addresses the reviewer's concern regarding unquantified errors in extrapolated cities.

[revised manuscript text omitted]

Please, clarify the state of calibration in the abstract and conclusions, explicating the emission models for 18 of the 20 cities utilize uncalibrated, extrapolated models.
Added as suggested.
**L17 – L19:** These models were calibrated using independent traffic observations available for Paris and Berlin, and subsequently applied to the remaining 18 cities in an extrapolated manner due to data availability constraints.

**L526 – L529** Several sources of uncertainty remain in our approach. Because the GPS-to-volume

conversion models were calibrated using in-situ sensor data from Paris and Berlin only and extrapolated to the remaining 18 cities, the results may be better suited for analysing spatial patterns, temporal dynamics, and relative differences across cities, rather than for precise reporting of absolute emission magnitudes.

There are also other related-concerns:

Comparison with Carbon Monitor (Figure 6) shows moderate correlations (R = 0.58-0.84) but systematic differences are not adequately explained. It is important to note that CM-city estimates are also based, in part, on consumer-driven mobility data like TomTom GPS. While Floating Car Data (FCD), such as that from TomTom, is valuable, it introduces significant discrepancies when compared against local traffic flow, as noted in previous literature (e.g.,doi:10.1002/essoar.10504783.1, doi:10.5194/egusphere-egu21-5419). The large discrepancies observed here warrant a much deeper investigation than a simple attribution to general "methodological differences.

The large discrepancies when comparing annual estimates with other high-resolution studies and city-specific inventories (ranging from −94% to −8%, and showing a ∼80% difference in Table S7) are concerning and must be better explained. The authors need to explain the major differences compared to the following studies: Ulrich et al., 2023 (-8.1% - low difference), Anjos et al., 2025 (-66 %) and Kühbacher et al.,2023 (-74.2%). It is important to note that the studies by Kühbacher et al. (which uses a bottom-up traffic model like VISUM) and Anjos et al. (which uses an ML-based bottom-up approach) both rely on local traffic counts from monitoring stations for their volume inputs.

My question is: What factor (s) is (are) limiting the $CO_2$ emission estimates derived from the FCD-based ML model? Given the inherent discrepancies in FCD when estimating actual traffic volume, is there a systematic bias in the GPS to volume conversion that consistently leads to the underestimation of emissions compared to inventories and studies that are anchored to local traffic counts?

As the difference is defined as (Other datasets / this study -1) * 100%, the negative differences indicate that our estimates are generally higher, not lower, than those of several reference studies.

At a conceptual level, total emissions inferred from fuel consumption statistics are often considered among the most accurate estimates at aggregated scales. However, city-level fuel consumption data are difficult to obtain and top-down spatial allocation (typically based on population or similar proxies) can introduce substantial uncertainty in attributing emissions to individual cities. These limitations partly explain the discrepancies observed among different city-scale emission datasets. Against this background, the relatively higher estimates obtained in this study mainly arise from two factors. First, our framework includes a more comprehensive set of road segments, particularly middle and small roads that are often partially or entirely excluded from other city-scale inventories (Figure 2). Second, as shown in our uncertainty analysis, emissions associated with small roads exhibit the largest variability and may be overestimated in data-scarce contexts.

Due to the lack of independent traffic counts for middle and small roads in most cities, it is hard to conclusively attribute the observed differences to systematic overestimation in our model. In our

framework, emission factors are comparatively well constrained, while the dominant source of uncertainty arises from GPS-based traffic volume estimation. Addressing this source of uncertainty therefore represents a key priority for future work. We have clarified these points in the revised Section 3.4 and Discussion.

3____

The choice to simply adopt an 80% training and 20% testing split can be quite simple in the context of ML. This "naive" splitting method can not be fully minimize overfitting or ensure the model is robust and generalizable to new, unseen data (data outside this study's scope). Why didn't you consider the validation techniques such as k-fold cross-validation, chronological splitting for time series data, or bootstrapping? These techniques are widely used to evaluate both the gap-filling model and the GPS-to-volume conversion ML model.

As suggested, we used k-fold cross-validation for both the gap-filling model and the GPS-to-volume conversion model. The detailed results are listed in **Table S3 and S6**. Because Table S3 is too long, here we put Table S6 as an example:

Table S6 Five-fold cross-validation performance of LightGBM for converting FCD sample counts to traffic volumes

| City | Road | Vehicle | R2_mean | R2_std | RMSE_mean | RMSE_std | MAE_mean | MAE_std |
|---|---|---|---|---|---|---|---|---|
| Paris | major | Car | 0.908 | 0.000 | 382.838 | 0.175 | 212.712 | 0.147 |
| Paris | middle | Car | 0.339 | 0.005 | 160.337 | 1.555 | 80.005 | 0.158 |
| Paris | small | Car | 0.344 | 0.010 | 122.352 | 2.638 | 67.067 | 0.196 |
| Paris | major | Truck | 0.882 | 0.000 | 127.949 | 0.100 | 70.726 | 0.061 |
| Paris | middle | Truck | 0.388 | 0.007 | 75.867 | 1.130 | 38.768 | 0.102 |
| Paris | small | Truck | 0.244 | 0.006 | 85.220 | 0.896 | 39.062 | 0.109 |
| Berlin | major | Car | 0.665 | 0.003 | 202.961 | 1.215 | 123.296 | 0.603 |
| Berlin | middle | Car | 0.856 | 0.002 | 137.126 | 1.162 | 89.794 | 0.603 |
| Berlin | small | Car | 0.880 | 0.002 | 143.225 | 1.106 | 83.985 | 0.940 |

Furthermore, while R2 is a good metric for assessing fit, it doesn't measure the error magnitude itself. To provide a complete picture of model performance, you should include RMSE and MAE from Table S2 directly in the main text.

We agree that $R^2$ alone does not fully characterize model performance and that error-based metrics such as RMSE and MAE are necessary to quantify the magnitude of prediction errors, but Table S2 is too long to put in the main text. To address this, we have added a concise summary of RMSE and MAE values to the main text (Table 3), reporting representative ranges (and mean values) across cities and road classes. The full city-level RMSE and MAE results remain provided in Table S2 for completeness.

**Table 3. Summary of LightGBM validation performance across cities and road classes.**

| Vehicle | item | Road class | Mean R² | Mean RMSE | Mean MAE |
|---------|------|-----------|---------|-----------|----------|
| Car | COUNT | Major | 0.93 | 16.34 | 9.08 |
| Car | COUNT | Middle | 0.73 | 6.09 | 3.91 |
| Car | COUNT | Small | 0.60 | 3.66 | 2.15 |
| Truck | COUNT | Major | 0.78 | 3.31 | 2.00 |
| Truck | COUNT | Middle | 0.57 | 1.88 | 1.29 |
| Truck | COUNT | Small | 0.54 | 1.87 | 1.15 |
| Car | SPEED | Major | 0.89 | 6.72 | 4.64 |
| Car | SPEED | Middle | 0.67 | 6.71 | 4.87 |
| Car | SPEED | Small | 0.58 | 7.85 | 5.63 |
| Truck | SPEED | Major | 0.84 | 8.77 | 6.35 |
| Truck | SPEED | Middle | 0.55 | 7.81 | 5.85 |
| Truck | SPEED | Small | 0.56 | 7.70 | 5.65 |

Table 2 lists eight features, but the justification for choosing these specific features is limited. Please provide a clearer explanation, including a literature basis, for why these predictors were selected over others. Since more potential predictors can often improve ML model performance, testing different feature engineering approaches for different road classes could be addressed for future research.

We thank the reviewer for this helpful comment. Our feature selection follows the framework of Xavier et al., who showed that road attributes and traffic activity variables are effective predictors for high-resolution emission estimation. Given our goal to develop a scalable approach applicable across multiple European cities, we deliberately used a compact set of predictors that are consistently available and computationally efficient.

High-resolution data availability also constrained the choice of features. Many potentially useful variables—such as traffic signals, fleet composition, or detailed built-environment metrics—are not uniformly accessible at the spatial resolution required. Since low-resolution predictors contribute little at a 100-m grid scale, we focused on variables with reliable, high-resolution coverage.
We agree that additional feature engineering could further improve model performance. As part of our ongoing work, we are considering incorporating high-resolution building-type information to enhance predictions, especially across different road classes. Our framework allows us to add more features easily for updating the dataset.

We Added **L571-L574**: As part of ongoing work, we plan to incorporate high-resolution urban context information, such as building-type data, to better capture heterogeneity across different road classes. The proposed framework is flexible and allows additional features to be integrated as new data becomes available.

4___

It's great that the data is available on Zenodo. But, the code is not mentioned. Since the Python code is integral to your entire methodology—covering all steps, from pre-processing and spatial operations to training, prediction, and CO2 emission calculation—I strongly recommend depositing it on GitHub and Zenodo. Providing well-documented code and samples is necessary for transparency and reproducibility.

We thank the reviewer for emphasizing the importance of code availability and fully agree that sharing code is essential for transparency and reproducibility. However, the Python code used in this study cannot be publicly released because it is tightly coupled with proprietary and confidential input data, and releasing it would risk exposing sensitive data structures. We have clarified this limitation in the Data and Code Availability section and provide a detailed methodological description in the manuscript, while all derived emission datasets are made openly available on Zenodo.

5____

The paper states "we used $EFCO_2$ of the EU6 standard" because "$CO_2$ emission factors are only marginally influenced by emission standards" (p.10). I think is imprecise due to Euro standards primarily target air pollutants, but $CO_2$ varies also significantly by vehicle age/technology, for instance.

We agree that Euro emission standards primarily regulate air pollutants (e.g., NOx, CO, PM), while $CO_2$ emissions are more strongly linked to engine efficiency, vehicle size, and age. However, we followed the COPERT methodology, where $CO_2$ emission factors are parameterized by vehicle category and technology classes that are defined consistently with Euro standards; these technology classes implicitly reflect typical age and efficiency characteristics of the fleet (E1–E6, etc.). Therefore, using $EFCO_2$ corresponding to Euro 6 effectively incorporates the assumed age/technology structure embedded in COPERT.

To clarify and empirically assess the potential impact of accounting for Euro standards on $CO_2$ emission factors, we conducted an additional sensitivity analysis comparing $EF_{CO_2}$ values derived with and without explicitly accounting for Euro standards. The results are presented in the following figure. As shown in the figure, the differences in $EFCO_2$ at 50 km/h between the two approaches are relatively small across the examined cities and remain within a typical uncertainty range of $\pm7\%$. This indicates that, at the aggregated level considered in this study, explicitly accounting for Euro standards leads to only marginal changes in $CO_2$ emission factors compared to other dominant sources of uncertainty.

[Figure]

Figure: Comparison of $CO_2$ emission factors with and without accounting for Euro emission standards

The final urban CO2 emissions are currently based on 100x100 meter grid cells. For better comparability and utility in urban modeling, the emissions data, in general, are calculated and expressed as a density per unit area within those grid cells. Please, check if it is more suitable reporting the CO2 emissions in units of mass per area per time (e.g., CO2 m2 ) rather than in units derived from line-segment meters.

The line-based emissions are only used as an intermediate step before aggregating to grid cells. The final dataset is already provided as emissions per $100 \times 100$ m grid cell per hour, so the unit are already expressed as mass per area per time: g $CO_2$ / per grid($10000$ m$^2$).

6__

While uncertainties are discussed qualitatively, no quantitative uncertainty estimates are provided for the emission maps. Note that: ML models have associated $R^2$ values, but these are not propagated to final emission uncertainties; GPS data coverage varies dramatically depending on road class, but impact on final emissions is not quantified. All of these limitations lead to a cumulative effect of multiple uncertainty sources that is unknown and unreported in the text.

It will be useful, at least, provide uncertainty ranges for annual city emissions (e.g., Berlin: $7.9 \pm X$ Mt $CO_2$,    Paris: $1,94 \pm X$ Mt $CO_2$). Please, be carful to clearly distinguish between precision (model $R^2$) and accuracy (comparison with true emissions)

As explained in our response to the first comment, quantitative uncertainty ranges for annual city-scale $CO_2$ emissions are now provided using a Monte Carlo–based uncertainty propagation framework. This approach explicitly accounts for traffic volume uncertainties across road classes and varying GPS data coverage, and yields 95% confidence intervals for each city (Figure 10 and Figure S9).

7__

The limitations are well-discussed in Section 4 but should be elevated, in my opinion. Maybe adding brief limitations statement to abstract, creating a "Limitations" subsection in discussion, and quantifying limitations wherever possible (don't just say "may lead to bias").

As suggested, we re-divided the discussion and updated **"Limitations" part(L526 – L565)**:

[revised manuscript text omitted]